# Flexibility-Aware Geometric Latent Diffusion for Full-Atom Peptide Design

**Dongjiang Niu**[1]   **Xiaofeng Wang**[2]   **Zhiqiang Wei**[1 3 4]   **Zhen Li**[1 5]

## Abstract

Although peptides are well suited for flexible and shallow binding interfaces, their intrinsic flexibility induces a strongly coupled sequence–structure relationship that current fixed-geometry latent models cannot simultaneously model with conformational diversity and physical feasibility, ultimately limiting design quality. To overcome this bottleneck, PepFGLD is proposed as a receptor-conditioned, flexibility-aware framework for full-atom peptide design. The framework is motivated by a systematic analysis of existing limitations: geometry shifts driven by interfacial flexibility are not well captured by standard equivariant encoders; the static combination of sequence information and 3D geometry cannot represent their dynamic interactions; and diffusion models without timely geometric feedback tend to drift away from physically reasonable energy landscapes. In PepFGLD, FlexEGNN is used to improve the sensitivity of geometric representations to local flexibility, a coherent and adaptable latent conformational manifold is formed through bidirectional sequence–structure interaction and nonlinear latent mapping, and a time-dependent energy-guided diffusion mechanism is incorporated to balance exploration and convergence during diffusion so that sampling trajectories are continuously guided toward physically feasible full-atom structures. PepFGLD yields improved binding affinity and design success across multiple peptide design tasks.

[1]College of Computer Science and Technology, Qingdao University, Qingdao, China [2]MindRank AI Ltd, Hangzhou, China [3]College of Computer Science and Technology, Ocean University of China, Qingdao, China [4]Shandong Provincial Key Laboratory of Intelligent Molecular Science and Engineering, Qingdao, China [5]Shandong Provincial Key Laboratory of Pathogenesis and Prevention of Brain Diseases, Qingdao, China. Correspondence to: Zhen Li <lizhen@qdu.edu.cn>.

*Proceedings of the 43rd International Conference on Machine Learning*, Seoul, South Korea. PMLR 306, 2026. Copyright 2026 by the author(s).

## 1. Introduction

Peptides are flexible biomolecules composed of a small number of amino acids and are capable of performing precise recognition and regulatory functions in diverse biological processes(Fosgerau & Hoffmann, 2015). As their value in modulating protein-protein in teractions, constructing targeted binding molecules, and enabling new therapeutic modalities has become increasingly evident, the need to design peptides with high specificity for given receptors has grown rapidly(Lau & Dunn, 2018). The central challenge, however, lies in generating peptides that can achieve stable and selective binding on complex receptor interfaces(Ciemny et al., 2018).

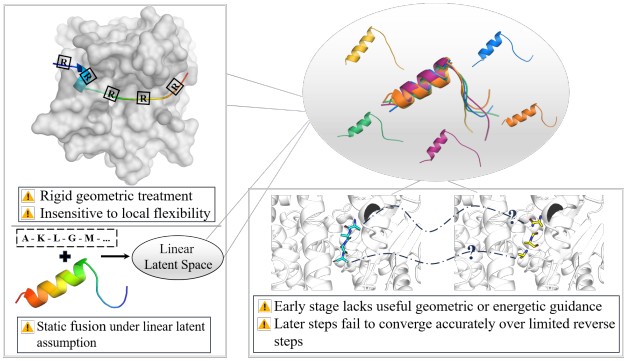

*Figure 1.* Illustration of peptide flexibility.

The development of generative models has increasingly become a key technical route for advancing peptide design. In recent years, diffusion-based generative frameworks have achieved rapid progress in molecular, protein, and three-dimensional structure generation, demonstrating strong capabilities in structural representation and sampling(Watson et al., 2023; Corso et al., 2022). Existing diffusion-based approaches fall into two categories. The first category performs noise injection and denoising directly in the original structural space, applying diffusion to atomic coordinates or backbones(Hoogeboom et al., 2022). Although effective for learning structural distributions, these methods often face instability and high computational cost when handling high-dimensional data, flexible conformations, and atom-level geometric constraints(Xu et al., 2022). The second category maps high-dimensional or variable-length structures into a continuous latent space using an encoder, performs

diffusion within this latent space, and then reconstructs the structure via a decoder(Rombach et al., 2022; Jin et al., 2018). While both categories have shown strong results in tasks such as protein generation(Watson et al., 2023), molecular generation(Xu et al., 2022), and fragment or linker design(Igashov et al., 2024), they remain insufficient for receptor-conditioned peptide generation, where high flexibility, receptor-driven constraints, and full-atom accuracy are required. As a result, these approaches cannot be directly applied to the generation of peptides tailored to a specific receptor.

However, as shown in Fig. 1, the existing peptide design framework still faces several key problems. First, peptides and their binding interfaces exhibit pronounced conformational flexibility and environment-dependent behavior, while commonly used equivariant structural encoders rely on fixed geometric processing schemes that are not well suited for capturing local flexibility, scale variations, or interface deformation. Second, peptide sequence semantics and geometry influence each other in a tight and complex manner, yet current sequence-structure joint models typically fuse the two modalities in a static way. When coupled with latent spaces that often assume linear priors, these models struggle to represent conformational changes. Finally, during diffusion-based generation, the absence of effective geometric or energetic guidance in high-noise stages can cause sampling trajectories to drift away from reasonable conformations. In later stages, as noise decreases, models often fail to converge to physically feasible structures within a limited number of reverse steps, leading to suboptimal geometric plausibility and local conformational quality. These limitations pose significant challenges for building a receptor-conditioned, full-atom peptide generation framework.

To this end, PepFGLD is proposed, a framework for Full-Atom Peptide design with Flexibility-Aware Geometric Latent Diffusion. FlexEGNN is introduced to provide equivariant structural encoding with explicit sensitivity to conformational flexibility, achieved through a learnable flexibility-modulation mechanism that adaptively adjusts interaction strength according to local flexibility and spatial scale, together with learnable channel modulation mechanism over multi-channel geometric representations to enable coordinated updates driven by sequence semantics and local conformational states. On this basis, a bidirectional sequence-structure interaction module (SSBIM) coupled with a nonlinear latent-variable kernel is employed to enhance the expressiveness of the latent space, allowing it to capture multimodal conformational distributions and form a more continuous and realistic conformational manifold. Furthermore, a time-dependent energy-guided diffusion (TDEG) mechanism is incorporated to balance structural exploration and physical convergence, where weak physical constraints promote diversity at high-noise stages and are progressively strengthened as noise decreases, guiding sampling trajectories toward energetically feasible full-atom conformations.

To sum up, our contributions can be summarized as follows:

- **Flexibility-aware equivariant structural encoding.** A Flexibility-Aware Equivariant Graph Neural Network (FlexEGNN) is introduced to incorporate conformational flexibility in full-atom peptide structures.

- **Bidirectional sequence–structure coupling in latent space.** A Bidirectional Sequence–Structure Interaction Module (SSBIM) is used to couple sequence semantics with structural representations in nonlinear latent space.

- **Time-dependent energy-guided diffusion process.** A Time-Dependent Energy-Guided Diffusion (TDEG) mechanism is adopted to progressively enforce physical constraints during sampling.

- **Experimental evaluation.** Experiments on peptide design benchmarks show that physically plausible full-atom conformations can be generated.

## 2. Related Work

In computational peptide design, force-field and docking pipelines remain common, exemplified by Rosetta Flex-PepDock for flexible peptide sampling and interface refinement(London et al., 2011). With deep learning, sequence-centric methods have been proposed, including CPL-Diff (diffusion, length-controlled functional peptides)(Luo et al., 2025) and AMP-Designer (LLM-based antimicrobial design)(Wang et al., 2025), while structure is explicitly modeled in GANDALF(Rossetto & Zhou, 2019) via separate GANs for sequence and structure. More recently, target-specific multimodal, structure-constrained design has been advanced by PepHAR(Li et al., 2024) and DiffPep-Builder(Wang et al., 2024), which couple sequence preferences with geometric constraints under receptor or hotspot conditioning.

Recently, geometric diffusion has attracted increasing attention, where diffusion is performed directly in Euclidean structural space under equivariant constraints. For small molecules, EDM(Hoogeboom et al., 2022) and GeoDiff(Xu et al., 2022) apply coordinate-space diffusion for 3D structure generation; for macromolecules, SMCDiff(Trippe et al., 2022) conducts diffusion in backbone-frame space for scaffolding/generation. For structure-conditioned docking, Diff-Dock(Corso et al., 2022) diffuses ligand poses, while Diff-BindFR(Zhu et al., 2024) further models flexible side chains for full-atom docking. For peptides, HYDRA(Vishva Saravanan et al., 2024) adopts a hybrid coordinate-space diffusion framework for affinity/stability optimization, and RAPi-

Dock(Zhao et al., 2025) applies diffusion directly to protein–peptide complex conformations for pose refinement.

An alternative to direct structural diffusion is latent diffusion, where variable-length structures are encoded into a continuous latent space and decoded back. GeoLDM(Xu et al., 2023) uses an equivariant autoencoder for molecular geometry; dyMEAN(Kong et al., 2023) encodes full-atom antibody-antigen complexes with a dynamic multi-channel equivariant graph network. For peptides, PepGLAD(Kong et al., 2024) applies a receptor-conditioned VAE with E(3)-equivariant latent diffusion and PepMimic(Kong et al., 2025) generates short interface-mimicking peptides in latent space.

However, most existing structural encoders are based on static equivariant geometry and only weakly capture peptide flexibility in receptor environments. Meanwhile, prevalent latent formulations are often weakly expressive, making sequence-structure integration and stable continuity under conformational perturbations difficult. In addition, diffusion is commonly performed without noise-stage-adaptive physical guidance, hindering an effective trade-off between exploration and physical feasibility along the sampling trajectory.

## 3. Preliminaries

In this chapter, the peptide design task and the related notations are formally defined. Each structural sample is represented as an full-atom geometric graph $G_a = (a_i)_{i=1}^{N_a}$, where $a_i$ denotes the conformation and attributes of the $i$-th atom-level node, and $N_a$ is the total number of atoms in the graph. To model structural constraints more efficiently, a channel-augmented residue graph, referred to as the channel graph $G_{aa} = (\mathcal{V}, \mathcal{E})$, is further defined. The node set $\mathcal{V} = \{(S_i, R_i)\}_{i=1}^{N_{aa}}$ consists of residue-level nodes, where $S_i$ indicates the amino acid type of the $i$-th residue, and each residue is represented as $R_i = \{r_{i,c} \in \mathbb{R}^3\}_{c=1}^C$, where $C$ denotes the number of channels determined by data distribution statistics. We set $C$ to 14, corresponding to a set of predefined atomic positions including backbone and side-chain atoms, with $\mathbb{R}^3$ representing the spatial coordinates. The edge set $\mathcal{E}$ encodes contact relations between amino acid residues. The peptide segment and the binding site within a structure are denoted as $G_{aa}^P$ and $G_{aa}^B$, respectively. These notations support the conditional generation process in peptide design, where the goal is to learn the conditional distribution $q(G_{aa}^P|G_{aa}^B)$.

## 4. Methodology

The overall architecture of PepFGLD is shown in Fig. 2 a. Structural samples are first encoded into latent point clouds through a flexibility-aware sequence-structure variational autoencoder (Flex-VAE, as shown in Fig. 2 b) built upon Flex-

EGNN and SSBIM, where learnable flexibility-modulation and channel-modulation mechanisms are employed to adapt geometric interactions and representation updates to local conformational states. A two-stage training strategy is then applied to the latent diffusion model (LDM) operating in this latent space. In the first stage, large-scale protein fragments are used for unsupervised pretraining, enabling the LDM to learn how to recover the original latent representation from arbitrary noise perturbations without relying on protein-peptide interaction information. In the second stage, the LDM is fine-tuned using real protein-peptide complexes, during which TDEG (as shown in Fig. 2 c) is incorporated to learn the conformational preferences, contact geometry patterns, and spatial constraints of peptide segments at binding sites.

### 4.1. Flex-VAE for Latent Point Clouds

Flex-VAE consists of an encoder and a decoder, and is used to map $G_{aa}^P$ into a flexibility-aware latent point cloud representation under the conditioning of $G_{aa}^B$, and then reconstruct the full-atom complex from this latent space. Given a channel-graph pair $(G_{aa}^P, G_{aa}^B)$ from a protein-peptide complex, a subset of amino acid residues in $G_{aa}^P$ is first randomly masked and their side-chain geometry is removed, producing a perturbed input $\widetilde{G_{aa}^P}$. The encoder takes $(\widetilde{G_{aa}^P}, G_{aa}^B)$ as input and outputs a latent state $\{(H_i, Z_i)|i \in G_{aa}^P\}$ for each peptide residue, where $H_i$ denotes the sequence-structure joint latent state, and $Z_i$ represents the coordinate latent state of that residue in latent space. The decoder then reconstructs the peptide residue types and full-atom geometry from this latent state under the condition of $G_{aa}^B$, producing $\widehat{G_{aa}^P}$.

To explicitly model the highly variable local conformations in protein-peptide complexes, a flexibility-aware multi-channel equivariant architecture, FlexEGNN, is designed. A joint semantic–geometric kernel is constructed by the encoder over multi-channel residue coordinates, and a learnable flexibility modulation is injected such that both edge messages and coordinate updates are allowed to adapt to local conformational changes while E(3) equivariance is preserved.

First, a joint geometric kernel is constructed, for any neighboring residues $i$ and $j$, let their multi-channel coordinate sets be $R_i = (r_{i,c_i})_{c_i=1}^C$ and $R_j = (r_{j,c_j})_{c_j=1}^C$, with channel weights $w_{i,c_i}$ and $w_{j,c_j}$, which are learnable residue-specific gating coefficients indicating the relative importance of each geometric channel under the current conformational state. A compact edge-level geometric descriptor $g_{ij}$ is defined by kernelizing all channel-pair displacements and integrating channel semantics:

$$g_{ij} = \phi_r \left( \frac{vec\big(U_i^T(w_{i,c_i}||r_{i,c_i}-r_{j,c_j}||_2 w_{j,c_j})U_j\big)}{||vec(U_i^T(w_{i,c_i}||r_{i,c_i}-r_{j,c_j}||_2 w_{j,c_j})U_j)||_2+\zeta} \right) \quad (1)$$

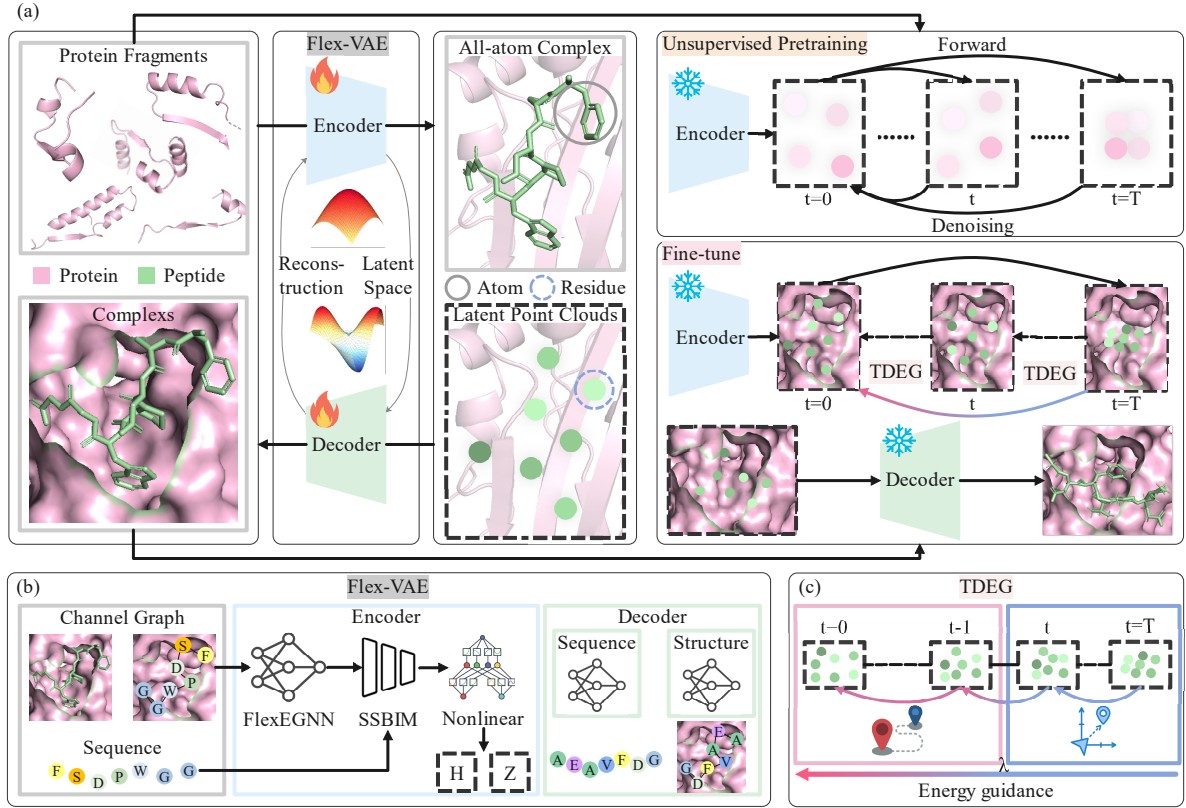

*Figure 2.* (a) Overall architecture of PepFGLD. (b) The detail of Flex-VAE. (c) The detail of TDEG.

where, $\phi_r$ denotes a learnable projection function, $U_i$ and $U_j$ are channel semantic embedding matrices, $vec$ denotes the matrix vectorization operation, and $\zeta$ is a bias term that ensures numerical stability, $\|\|_2$ denotes the 3D Euclidean distance. This construction jointly captures spatial scale, channel-level semantic interactions, and local conformational flexibility in a differentiable form, enabling Flex-EGNN to distinguish geometrically similar yet flexibly distinct neighborhoods without breaking equivariance.

Next, a flexibility-modulation kernel is introduced to explicitly capture variations in residue flexibility along the sequence. Let $p_i$ and $p_j$ be sequence residue indices and $\Delta_{i,j} = |p_i - p_j|$. After discretizing $\Delta_{i,j}$ into finite levels, learnable modulation parameters $(\omega_\Delta, b_\Delta)$ per level is maintained and the edge message is modulated accordingly:

$$\hat{h}_i = \psi_h\left(h_i, \sum_{j \in \mathcal{N}(i)} \omega_{\Delta_{i,j}} \phi_e\left(h_i\|h_j\|g_{ij}\right) + b_{\Delta_{i,j}}\right) \quad (2)$$

where $h_i$ and $h_j$ are node features, $\phi_e$ is an edge transformation, $\|$ denotes concatenation, $\psi_h$ is the node update mapping, $\mathcal{N}(i)$ denotes the set of neighboring residues $j$. The flexibility-modulation kernel allows the same geometric pattern to exert different levels of influence across distinct conformational or flexibility regions, thereby enabling the encoder to explicitly sense local flexibility.

FlexEGNN adopts an E(3)-equivariant formulation for co-

ordinate updates. First, channel-level response coefficients are computed from the modulated messages, and the final equivariant channel weights are obtained through learnable channel-modulation mechanism:

$$\widetilde{\alpha}_{ij,c} = \frac{exp\left(\phi_c\left(\omega_{\Delta_{i,j}} \phi_e(h_{i,c}\|h_{j,c}\|g_{ij,c}) + b_{\Delta_{i,j}}\right)\right)}{\sum_{k \in \mathcal{N}(i)} exp\left(\phi_c\left(\omega_{\Delta_{i,k}} \phi_e(h_{i,c}\|h_{k,c}\|g_{ik,c}) + b_{\Delta_{i,j}}\right)\right)} \quad (3)$$

where $\phi_c$ denotes the learnable channel fusion mapping. To avoid dependence on any global frame, we use channel-weighted centers as references:

$$z_i = \sum_{c=1}^{C} \frac{w_{i,c}}{\sum_{c'=1}^{C} w_{i,c'}} r_{i,c}, \bar{z}_j = \sum_{c=1}^{C} \frac{w_{j,c}}{\sum_{c'=1}^{C} w_{j,c'}} r_{j,c} \quad (4)$$

Under the current conformational environment, the flexibility coupling strength is defined and used to update the latent point-cloud coordinates as follows:

$$\hat{z}_i = z_i + \frac{1}{|\mathcal{N}(i)|} \sum_{j \in \mathcal{N}(i)} \widetilde{\alpha}_{ij} \sum_{c=1}^{C} \frac{w_{i,c}}{\sum_{c'=1}^{C} w_{i,c'}} g_{ij,c}(z_i - \bar{z}_j) \quad (5)$$

This update is constructed entirely from relative displacements, thereby maintaining strict equivariance under rotation and translation, a formal proof is provided in Appendix A. Meanwhile, the learned coupling $k_{ij}$ allows the latent point clouds to adaptively bend and flow with local conformational states, providing a flexibility-consistent coordinate latent for downstream reconstruction.

FlexEGNN is operated primarily over structural neighborhoods, while sequence information is only injected as an initial node attribute; consequently, the interaction between sequence semantics and geometric semantics is weakened. To align residue types with their geometric context at the node level, SSBIM is introduced, in which mirrored cross-modal attention operators are applied so that geometric information is propagated into the sequence stream, while sequence semantics are simultaneously used to modulate the geometric stream:

$$\hat{S}_i = S_i + \mu \sum_{j \in G_{aa}^P} \frac{exp(<S_i W_q^s, \hat{h}_j W_k^h>/\sqrt{d_h})}{\sum_{n \in G_{aa}^P} exp(<S_i W_q^s, \hat{h}_n W_k^h>/\sqrt{d_h})} \hat{h}_j \quad (6)$$

$$\hat{\hat{h}}_i = \hat{h}_i + \vartheta \sum_{j \in G_{aa}^P} \frac{exp(<\hat{h}_i W_q^h, S_j W_k^s>/\sqrt{d_h})}{\sum_{n \in G_{aa}^P} exp(<\hat{h}_i W_q^h, S_n W_k^s>/\sqrt{d_h})} S_j \quad (7)$$

where scalar gating coefficients $\mu$ and $\vartheta$ are trainable parameters, $W_q^s$, $W_k^h$, $W_q^h$, and $W_k^s$ are learnable projection matrices, and $<>$ denotes the inner product. To obtain sequence and geometric latent variables that are suitable for continuous modeling in latent space, a kernel-based nonlinear mapping is introduced into the latent point clouds(Liu et al., 2024). For $H_i$, the Gaussian posterior is parameterized from $\hat{h}_i$ by a radial-basis-function activation network $\psi_H$, in which global trends and local sensitivities are jointly modeled:

$$(\mu_i^h, \log \sigma_i^h) = \psi_H(\hat{\hat{h}}_i) = W_b^h \varphi(\hat{\hat{h}}_i) + W_s^h R(\hat{\hat{h}}_i; \{c_m\}_{m=1}^M)$$
$$q_\psi(H_i \mid G_{aa}^P, G_{aa}^B) = \mathcal{N}(H_i; \mu_i^h, \text{diag}((\sigma_i^h)^2)) \quad (8)$$
$$H_i = \mu_i^h + \exp(\tfrac{1}{2} \log \sigma_i^h) \odot \varepsilon_i^h, \quad \varepsilon_i^h \sim \mathcal{N}(0, I)$$

where $\varphi$ denotes a smooth nonlinear mapping, and $R(\hat{h}_i; \{c_m\}_{m=1}^M)$ is an expansion over radial basis functions with $M$ reference points $\{c_m\}_{m=1}^M$ uniformly placed in feature space. $W_b^h$ and $W_s^h$ are learnable mapping matrices, and $\sigma_i^h = exp(\tfrac{1}{2} log\sigma_i^h)$ is induced.

For $Z_i$, deviations of channel-level geometry with respect to the backbone anchor are modeled. Let $\bar{\bar{z}}_i$ denote the channel anchor coordinate of residue $i$, and define $\Delta_{z_i} = z_i - \bar{\bar{z}}_i$. A zero-offset prior is imposed and the posterior of $\Delta_{z_i}$ is parameterized by a kernel activation network $\psi_Z$ from $\hat{\hat{h}}_i$:

$$q_\psi(\Delta_{z_i}|G_{aa}^P, G_{aa}^B) = \mathcal{N}(\Delta_{z_i}; 0, diag((\sigma_i^z = exp(\tfrac{1}{2}\psi_Z(\hat{\hat{h}}_i)))^2)) \quad (9)$$
$$Z_i = \bar{\bar{z}}_i + \sigma_i^z \odot \varepsilon_i^z, \ \varepsilon_i^z \ \mathcal{N}(0, I)$$

For the decoder, a structure is used that is isomorphic to FlexEGNN to jointly model the peptide and the binding site. It reconstructs the peptide residue types and full-atom coordinates from the latent point clouds, producing $\{(\hat{H}_i, \hat{Z}_i)|i \in \widehat{G_{aa}^P}\}$.

During training, a masked residue set $\Omega_S$ is used for sequence supervision, while structural reconstruction is evaluated over all valid atoms $\Omega_{atom}$ and additionally constrained over the backbone $C_\alpha$ set $\Omega_{CA}$. KL regularization is applied to both $H_i$ and $\Delta_{z_i}$, with respect to standard isotropic Gaussian priors, and the overall objective is written as:

$$\mathcal{L}_{VAE} = (1 - \rho)\mathcal{L}_{seq} + \rho\mathcal{L}_{str} + \lambda_h \mathcal{L}_{KL}^h + \lambda_z \mathcal{L}_{KL}^z \quad (10)$$

where $\rho$ controls the relative weight between sequence reconstruction and structural reconstruction, and $\lambda_h$ and $\lambda_z$ are weighting hyperparameters. $\mathcal{L}_{seq}$ denotes the masked residue-type reconstruction loss over $\Omega_S$, $\mathcal{L}_{str}$ denotes the structural reconstruction loss computed over $\Omega_{atom}$ with an additional backbone constraint over $\Omega_{CA}$, $\mathcal{L}_{KL}^h$ and $\mathcal{L}_{KL}^z$ denote the KL divergence that regularizes the posterior of $H_i$ and $\Delta_{z_i}$ toward $\mathcal{N}(0, I)$, respectively, with detailed computations provided in the Appendix B.

## 4.2. Geometric Latent Diffusion with TDEG

After obtaining the Flex-VAE encoded latent states $\{(H_i, Z_i)|i \in G_{aa}^P\}$, a diffusion model is constructed in this space. A two-stage training strategy is adopted. First, unsupervised pretraining is performed on large-scale native protein structural fragments, enabling the model to learn how to recover latent representations from arbitrary noise perturbations without relying on any protein–peptide interaction information. The model is then fine-tuned on real protein-peptide complex data, and a TDEG is introduced during sampling to capture conformational preferences, contact geometry patterns, and local physical constraints of peptides at binding interfaces.

First, forward diffusion noise is applied simultaneously to $H_i$ and $Z_i$, yielding a sequence of intermediate variables $(H_i^t, Z_i^t)$:

$$q(H_i^t|H_i) = \mathcal{N}(H_i^t; \sqrt{\alpha^t}H_i, (1 - \alpha^t)I)$$
$$q(Z_i^t|Z_i) = \mathcal{N}(Z_i^t; \sqrt{\alpha^t}Z_i, (1 - \alpha^t)I) \quad (11)$$

where, the noise schedule $\{\beta_t\}$ follows a cosine schedule, and $\alpha^t = \prod_{s=1}^t (1 - \beta_t)$. The reverse diffusion process is parameterized by a denoising network that jointly predicts sequence-structure noise and geometric noise, and the corresponding conditional distribution is defined as:

$$p_\theta(H_i^{t-1}, Z_i^{t-1}|H_i^t, Z_i^t, G_{aa}^B) = \mathcal{N}\left(\begin{bmatrix} H_i^{t-1} \\ Z_i^{t-1} \end{bmatrix}; \begin{bmatrix} \mu_\theta^h(H_i^t, Z_i^t, G_{aa}^B, t) \\ \mu_\theta^z(H_i^t, Z_i^t, G_{aa}^B, t) \end{bmatrix}, \beta_t I\right) \quad (12)$$

where $\mu_\theta^h$ and $\mu_\theta^z$ are jointly predicted by a unified denoising network conditioned on $(H_i, Z_i, P_i, G_{aa}^B)$. Here, $P_i$ denotes the absolute positional embedding of the $i$-th residue, enabling the coordinated recovery of the sequence-structure latent state and the geometric latent state.

To promote physically plausible geometries, TDEG is injected into the reverse trajectory by augmenting the coordinate update with a lightweight differentiable geometric energy $E_{phys}(Z)$ (the detail is presented in Appendix C),

which is composed of terms for consecutive residue distances, non-bonded collisions, local bond angles/dihedrals, and spatial distribution. A monotonically increasing schedule $\lambda(t)$ (the detail is presented in Appendix C) is applied so that this constraint is emphasized only as the noise level decreases, leading to the unified reverse update:

$$Z_i^{t-1} = Z_i^t - \beta_t \left[ \epsilon_\theta^z \left( H_i^t, Z_i^t, G_{aa}^B, t \right) + \lambda(t) \nabla_{Z_i} E_{phys}(Z) \right] \quad (13)$$

where $\nabla_{Z_i} E_{phys}(Z)$ denotes the gradient of the differentiable geometric energy with respect to the latent coordinates. A detailed theoretical analysis and formal justification are presented in Appendix D. In this way, diverse conformations are permitted at high-noise stages, while convergence toward feasible structures is progressively enforced at low-noise stages. During training, standard noise prediction supervision is retained:

$$\mathcal{L}_{LDM} = \mathbb{E}_{t \sim U(1,T)} \mathbb{E}_{(H_i, Z_i), \epsilon} \left[ ||\epsilon - \epsilon_\theta(H_i^t, Z_i^t, G_{aa}^B, t)||_2^2 \right] \quad (14)$$

where $\epsilon$ denotes the standard Gaussian noise added in the forward process, and $\epsilon_\theta$ is the noise predicted by the model.

# 5. Experiments

## 5.1. Setup

**Datasets:** For datasets, the PepBench, PepBDB, and Prot-Frag datasets prepared in PepGLAD(Kong et al., 2024) are adopted, with only a small number of entries that failed coordinate parsing removed from the original sample lists, which ensures high-quality input data while preserving the original data distribution. We show details of these datasets in Appendix E. **Tasks:** For peptide design, the models are evaluated under two scenarios. The first is the Sequence-Structure Co-Design task, in which both the peptide sequence and its conformation are generated simultaneously under the condition of a given receptor binding interface. The second is the Binding Conformation Generation task, where the peptide sequence is provided and the goal is to generate its spatial conformation and binding pose with respect to the target receptor binding site. **Baselines:** For baselines, since only a limited number of methods are able to support both peptide design tasks, five representative models are selected under the principles of methodological consistency and minimal modification, including HSRN(Jin et al., 2022), dyMEAN(Kong et al., 2023), GeoLDM(Xu et al., 2023), PepGLAD(Kong et al., 2024) and PepMimic(Kong et al., 2025). We show details of baselines in Appendix F. **Metrics:** For evaluation metrics, in the Sequence-Structure Co-Design task, binding energy $\Delta G$ (Rosetta(Alford et al., 2017)) Success rate, Diversity (Div), and Consistency (Con) are reported. In the Binding Conformation Generation task, RMSD($C_\alpha$), RMSD(atom) and DockQ are reported. We show details of metrics in Appendix G. We provide the overall training and sampling procedures in Algorithm 1 and 2 in

*Table 1.* Evaluation on sequence-structure co-design.

| Model | PepBench | | | | PepBDB | | | |
|---|---|---|---|---|---|---|---|---|
| | $\Delta G \downarrow$ | Success $\uparrow$ | Div $\uparrow$ | Con $\uparrow$ | $\Delta G \downarrow$ | Success $\uparrow$ | Div $\uparrow$ | Con $\uparrow$ |
| Reference | -37.73 | 97.84% | - | - | -36.32 | 97.36% | - | - |
| HSRN | $\geq 0$ | 10.46% | 0.158 | 0.0 | $\geq 0$ | 10.86% | 0.111 | 0.0 |
| dyMEAN | -2.26 | 14.60% | 0.150 | 0.0 | -1.92 | 6.26% | 0.150 | 0.0 |
| GeoLDM | -17.22 | 50.45% | 0.5128 | 0.7447 | -17.65 | 40.17% | 0.8087 | 0.9347 |
| PepGLAD | -13.46 | 47.12% | 0.5677 | 0.7648 | **-19.57** | 47.17% | 0.7985 | 0.9278 |
| PepMimic | -11.21 | 52.28% | 0.5527 | 0.7829 | -19.27 | 48.69% | 0.8039 | **0.9573** |
| PepFGLD | **-19.86** | **59.35%** | **0.5824** | **0.8160** | -18.74 | **49.67%** | **0.8276** | 0.9484 |

**Appendix L. Computation cost:** We compare the computational cost of PepFGLD and PepGLAD under the same environment. On an NVIDIA A100-40G GPU, when sampling 40 candidate structures for each test sample, PepFGLD requires 26.73 seconds per sample, while PepGLAD requires 19.91 seconds. The parameter count increases from 3.58M in PepGLAD to 4.07M in PepFGLD, corresponding to an increase of approximately 13.7%. These results indicate that PepFGLD introduces additional overhead but remains within the same order of magnitude as PepGLAD.

## 5.2. Sequence-Structure Co-Design

The primary objective of Sequence-Structure Co-Design is to generate peptides that can stably bind to a given receptor interface. Therefore, $\Delta G$ and Success are the most informative metrics for assessing generation quality. $\Delta G$ reflects conformational stability, while Success indicates whether geometric and physical constraints are satisfied. Specifically, for each sample, 40 candidates are generated, and the results are summarized in Table 1. Across both benchmark datasets, PepFGLD achieves highly competitive overall performance in the Sequence-Structure Co-Design task, with particularly strong results in terms of $\Delta G$ and Success, significantly outperforming baselines. This indicates a superior capability to generate peptides with physically feasible binding conformations. Traditional non-diffusion approaches are clearly limited in scenarios with high peptide conformational variability, as the generated conformations often fail to satisfy the geometric constraints required for stable binding, resulting in higher $\Delta G$ values and success rates generally below 15%. Existing diffusion-based methods improve diversity but still exhibit deficiencies in geometric stability and final energy levels. In contrast, PepFGLD achieves the best or second-best binding energies on both datasets, demonstrating its ability to consistently generate conformations with lower energy and more reasonable binding states. The advantage of PepFGLD primarily arises from its systematic modeling of peptide flexibility. By maintaining sensitivity to local conformational variations, the model is able to more accurately describe the flexible adaptation of peptide chains near receptor interfaces. During encoding, local conformational adjustments induced by the receptor environment are captured and preserved in the representation; in latent space, these variations are further expressed in a continuous and physically consistent manner through nonlinear latent mappings; during diffusion, progressively strengthened physical

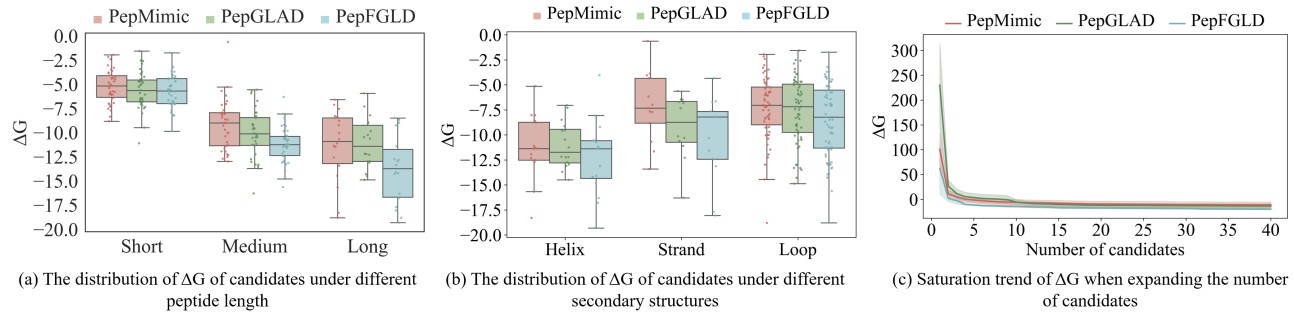

(a) The distribution of ΔG of candidates under different peptide length

(b) The distribution of ΔG of candidates under different secondary structures

(c) Saturation trend of ΔG when expanding the number of candidates

*Figure 3.* ΔG of models under different peptide characteristics on PepBench.

guidance ensures stable convergence of generated structures. This flexibility-aware modeling across multiple stages enables PepFGLD to preserve reasonable diversity while steering generation toward conformational regions that are both lower in energy and more geometrically feasible. An ablation study on the effectiveness of PepFGLD is provided in the Appendix H.

### 5.3. Binding Conformation Generation

*Table 2.* Evaluation on Binding Conformation Generation.

| Model | PepBench | | | PepBDB | | |
|---|---|---|---|---|---|---|
| | $RMSD_\alpha \downarrow$ | $RMSD_a \downarrow$ | DockQ ↑ | $RMSD_\alpha \downarrow$ | $RMSD_a \downarrow$ | DockQ ↑ |
| HSRN | 6.02 | 7.59 | 0.508 | 9.28 | 9.72 | **0.394** |
| dyMEAN | 7.96 | 8.35 | 0.374 | 17.64 | 17.56 | 0.142 |
| GeoLDM | 65.67 | 64.99 | 0.013 | **7.85** | 9.52 | 0.375 |
| PepGLAD | 4.45 | 5.63 | 0.558 | 11.02 | 11.99 | 0.290 |
| PepMimic | 5.12 | 6.33 | 0.520 | 8.41 | 9.49 | 0.371 |
| PepFGLD | **4.14** | **5.41** | **0.607** | 8.86 | **9.45** | 0.345 |

The goal of Binding Conformation Generation is to recover the binding conformation of a peptide under the conditions of a given sequence and receptor binding site. Specifically, 10 candidates are sampled for each example, and the results are reported in Table 2. GeoLDM performs poorly on the cross-target generalization dataset Pepbench, but demonstrates competitive performance on the homologous test set PepBDB. On PepBench, PepFGLD achieves the best performance across all three metrics, indicating strong in-distribution conformational reconstruction capability. Compared with non-diffusion methods, PepFGLD significantly reduces both backbone and full-atom errors, and relative to existing diffusion-based approaches, it further improves DockQ, suggesting that a more stable geometric representation is formed under receptor constraints. On the more structurally diverse and interface-complex PepBDB dataset, the performance differences among models become more pronounced. Although PepFGLD attains only second-best results in backbone RMSD and DockQ, indicating room for improvement in capturing global peptide positioning and interface contact patterns under large receptor shape variation and more complex interface topology, it is noteworthy that PepFGLD still achieves the best full-atom RMSD. This suggests that, even in highly variable scenarios, PepFGLD maintains stable full-atom quality, highlighting the advan-

tage of its flexibility-aware latent diffusion framework in terms of conformational physical plausibility.

### 5.4. Performance on Different Characteristics

To further evaluate the robustness of PepFGLD under varying peptide properties and sampling conditions, a set of extended experiments is conducted on PepBench for the Sequence–Structure Co-Design task. As shown in Figs. 3 and S1 Across different peptide lengths, secondary structure types, and sampling budgets, PepFGLD consistently achieves lower binding energies and improved geometric accuracy compared with strong baseline methods, indicating stable performance across diverse conformational regimes. Notably, PepFGLD demonstrates enhanced adaptability in scenarios involving larger conformational spaces or higher flexibility, while also reaching high-quality solutions with fewer sampled candidates. Detailed experimental settings and quantitative analyses are provided in the Appendix I.

### 5.5. Evaluation of Binding-Energy Fidelity

*Table 3.* Quantitative comparison of best candidate ΔG predictions across models on PepBench.

| Model | Spearman-$\rho$ | p(Spearman) | RMSE | t-stat | p(t) |
|---|---|---|---|---|---|
| PepMimic | 0.4500 | 6.0055 | 76.22 | 3.5611 | 5.8e-04 |
| PepGLAD | 0.5365 | 2.9619 | 60.47 | 4.2028 | 6.0e-05 |
| PepFGLD | **0.5981** | **2.4291** | **32.53** | **6.3031** | **9.9e-09** |

To evaluate the ability of the models to approach reference conformations within the binding-energy landscape, 40 candidate structures are independently sampled for each Pep-Bench example, and the conformation with the lowest ΔG is selected as the model's best candidate. The ΔG of the best candidate is then compared with the ΔG of the reference structure on a per-sample basis. The binding energy distributions of all samples generated by PepFGLD and the baselines are summarized in the Appendix J. Table 3 summarizes the quantitative results of the three models in terms of Spearman correlation, RMSE, and paired t-test statistics, with detailed definitions of these metrics provided in the Appendix J. Overall, PepFGLD achieves the best performance across all metrics. Specifically, PepFGLD attains the highest Spearman-$\rho$, indicating that it most effectively preserves

the relative energy ranking among different samples and can reliably reflect differences in conformational quality across receptor environments. Its RMSE is substantially lower than that of PepGLAD and PepMimic, suggesting that the energy distribution of the best candidates generated by PepFGLD is more concentrated with smaller deviations, and is less prone to producing abnormally high-energy conformations. In the paired t-test, PepFGLD yields the largest absolute t-statistic with the strongest statistical significance, indicating that the energies of conformations generated by PepFGLD are overall significantly lower than the baselines and thus exhibit stronger binding stability. Taken together, these results demonstrate that PepFGLD can more reliably approximate the reference energy landscape and generate low-energy candidate conformations that are more physically plausible.

## 5.6. Evaluator Alignment Analysis

*Table 4.* Additional evaluation using Prime MM-GBSA. Lower binding energy indicates more favorable binding.

| Method | Success Rate ↑ | Binding Energy ↓ |
|---|---|---|
| PepGLAD | 0.8508 | -99.73 |
| PepFGLD | **0.8758** | **-102.48** |

Since the TDEG in PepFGLD is based on differentiable geometric and physical priors, an additional evaluation is conducted to examine whether the observed improvement is tied to the PyRosetta-based evaluator, and we have therefore added a new experiment using Schrödinger Prime MM-GBSA. Specifically, we evaluated the generated protein-peptide complexes from PepFGLD and PepGLAD using Prime MM-GBSA. Prime MM-GBSA is a widely used post hoc scoring method that estimates the energetic favorability of a given bound complex under an alternative physics-based scoring framework. Its score can be viewed as an approximate proxy for relative binding free energy, where lower (more negative) values generally indicate more favorable binding. We did not use re-docking here because the target of evaluation is already a bound complex; re-docking would introduce an additional pose-sampling step and could alter the generated peptide conformation, making the result dependent on the docking procedure rather than on the generated complex itself. In contrast, MM-GBSA directly evaluates the generated bound state. We also note that Prime MM-GBSA here is used only for static post hoc scoring, not for short MD stability analysis. Experimental results demonstrate that PepFGLD achieves a higher candidate success rate (0.8758 vs. 0.8508) in terms of overall generation quality. Furthermore, when considering the best candidate structures among successful samples, PepFGLD yields a superior average binding energy (-102.48 kcal/mol vs. -99.73 kcal/mol). These findings remain consistent with the PyRosetta conclusions under an alternative scoring sys-

tem, suggesting that the performance improvement does not stem from alignment with a specific evaluator but reflects a genuine enhancement in structural quality and binding affinity.

## 5.7. Flexibility-aware Analysis

*Table 5.* Performance comparison on high-flexibility and low-flexibility peptide subsets.

| Method | Subset | $\Delta G \downarrow$ | Success ↑ |
|---|---|---|---|
| PepFGLD | High-Flex | **-14.54** | **53.00%** |
| | Low-Flex | **-27.58** | **68.55%** |
| PepGLAD | High-Flex | -7.11 | 43.18% |
| | Low-Flex | -22.66 | 52.83% |
| PepMimic | High-Flex | -3.90 | 46.59% |
| | Low-Flex | -21.79 | 60.53% |

To support the flexibility-aware claim, we use the proportion of coil (C) residues in the peptide secondary structure as a proxy for structural flexibility. This metric is selected because coil regions lack regular hydrogen-bonding patterns and are generally considered more flexible than regular secondary structures. Specifically, secondary structure assignments are obtained using the PSSpred module of I-TASSER. For each peptide sequence, predicted secondary structures involving helix (H), beta-sheet (E/B), and coil (C) are generated, and the fraction of coil residues is calculated to measure flexibility. Using this proxy, we partition the test set into High-Flex and Low-Flex subsets and evaluate the performance of PepFGLD alongside two representative baselines, PepGLAD and PepMimic. The results show that all methods exhibit worse $\Delta G$ on the High-Flex subset compared to the Low-Flex subset, which aligns with the classical interpretation of conformational entropy penalties(Chen et al., 2019). However, on this challenging subset, PepFGLD achieves $\Delta G$ improvements of 7.43 and 10.65 over the baselines, respectively, representing a significantly larger margin than the gains of 4.91 and 5.78 observed in the Low-Flex subset. Furthermore, PepFGLD is the only method that achieves a Success rate exceeding 50% in the High-Flex category.

## 5.8. Case Study

To assess peptide generation performance under receptor-imposed structural constraints, we conduct a case study on representative protein–peptide complexes. Two complexes (PDB: 6fq4(Whitewood et al., 2018) and 6qg8(Murray et al., 2019)) are selected to reflect distinct geometric and conformational challenges, with detailed selection rationales provided in Appendix K. Experimental results show that PepFGLD consistently achieves lower binding energies and more coherent interface conformations than baseline meth-

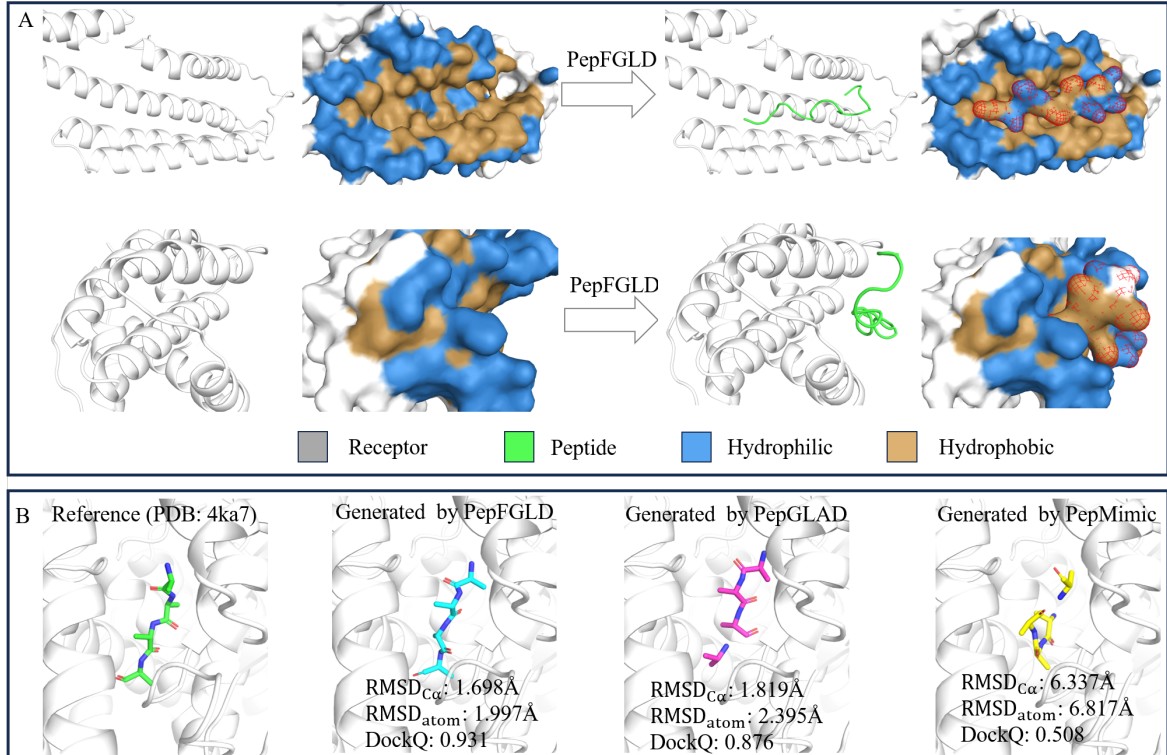

*Figure 4.* A.Top: A generated candidate confined within the binding site (PDB=6fq4, ΔG=-42.18). Bottom: A generated candidate with complementary shape to the binding site (PDB=6qg8, ΔG=-41.24). Both candidates form compact interactions at the interface. B. Left: Reference peptide structure of PDB 4KA7. Right: Generated peptides from PepFGLD, PepGLAD, and PepMimic with corresponding RMSD and DockQ scores.

ods, particularly under strong geometric confinement and in systems requiring large-scale conformational rearrangements, indicating superior flexibility modeling across both constrained and highly flexible binding scenarios, with in-depth analyses reported in Appendix K.

To qualitatively evaluate backbone stability and geometric continuity in peptide generation, we conduct a representative case study on a PepBench complex. The complex 4KA7(Kmiec et al., 2013) is selected due to its shallow groove geometry and high sensitivity to backbone perturbations, with detailed structural characteristics discussed in Appendix K. Results show that PepFGLD accurately reconstructs a continuous backbone closely matching the native conformation, maintaining coherent backbone trajectories under strong geometric constraints, whereas baseline methods exhibit fragmented structures, despite partially recovering global binding orientations, highlighting PepFGLD's advantage in preserving geometric coherence, as further analyzed in Appendix K.

## Conclusion

In this paper, PepFGLD is presented as a receptor-conditioned and flexibility-aware framework for full-atom peptide generation, motivated by the high flexibility of peptide–receptor interfaces and the nonlinear coupling between sequence semantics and geometry. By integrating flexibility-sensitive geometric encoding, bidirectional sequence–structure latent representations, and time-scheduled physical guidance within a latent diffusion framework, stable, physically consistent, and controllable conformational generation is achieved. Extensive evaluations across multiple benchmarks demonstrate consistent improvements in binding energy, success rate, and conformational accuracy across diverse peptide lengths, structural types, and sampling budgets. A remaining limitation is that receptor-side conformational dynamics are not explicitly modeled, which will be addressed in future extensions.

## Software and Data

Codes for PepFGLD as well as the experiments are available at https://github.com/NiuDongjiang/PepFGLD.

## Acknowledgments

This work were supported by National Natural Science Foundation of China (12371491) and Key Project of the Qingdao Natural Science Foundation (25-3-1-2-zyyd-jch).

## Impact Statement

This paper presents work whose goal is to advance the field of Machine Learning. There are many potential societal consequences of our work, none of which we feel must be specifically highlighted here.

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

# A. Theoretical Analysis of FlexEGNN

## A.1. Preliminaries and Notation

In FlexEGNN, each peptide residue $i \in G_{aa}^P$ carries a multi-channel coordinate set

$$R_i = (r_{i,c})_{c=1}^{C}, r_{i,c} \in \mathbb{R}^3 \tag{15}$$

with learnable channel weights $w_{i,c}$. Let $T \in E(3)$ denote an arbitrary global rigid transformation acting on coordinates by

$$T(r) = Rr + t, \; R \in SO(3), \; t \in \mathbb{R}^3 \tag{16}$$

We use $T(R_i)$ to denote applying $T()$ to each channel coordinate in $R_i$. Following the common equivariant-learning convention, a mapping that outputs 3D vectors is E(3)-equivariant if it commutes with $T$, while a mapping that outputs features is E(3)-invariant if its output is unchanged under $T$.

The FlexEGNN encoder constructs an edge-level geometric descriptor $g_{i,j}$, modulates edge messages by a sequence-distance dependent flexibility modulation $(w_{\Delta_{i,j}}, b_{\Delta_{i,j}})$ where $\Delta_{i,j} = |p_i - p_j|$, and updates latent coordinates using relative displacements between channel-weighted centers $z_i$ and $\bar{z}_j$.

The purpose is to establish that: (i) FlexEGNN remains strictly E(3)-equivariant despite flexibility modulation; (ii) flexibility modulation injects a symmetry-preserving, sequence-conditioned anisotropy rather than a frame-dependent geometric bias.

## A.2. Invariance of the Joint Geometric Kernel

### Definition A.2.1 (Edge-feature invariance)

Let $g_{ij}$ be the edge-level geometric descriptor defined in Eq. (1). We call $g_{ij}$ E(3)-invariant if for any rigid transformation $T \in E(3)$

$$g_{ij}(T(R_i), T(R_j)) = g_{ij}(R_i, R_j) \tag{17}$$

### Lemma A.2.2 (Relative displacement equivariance)

For any channels $c_i, c_j$ and any $T(r) = Rr + t$, the displacement transforms as

$$(r_{i,c_i} - r_{j,c_j}) \rightarrow R(r_{i,c_i} - r_{j,c_j}) \tag{18}$$

### Proof

$$T(r_{i,c_i}) - T(r_{j,c_j}) = (Rr_{i,c_i} + t) - (Rr_{j,c_j} + t) = R(r_{i,c_i} - r_{j,c_j}) \tag{19}$$

### Proposition A.2.3 (E(3)-invariance of $g_{ij}$)

The descriptor $g_{ij}$ in Eq. (1) is E(3)-invariant.

### Proof

Eq. (1) constructs $g_{ij}$ by kernelizing all channel-pair displacements via Euclidean norms $||r_{i,c_i} - r_{j,c_j}||_2$, combining them with channel semantics through $U_i, U_j$, applying $vec()$, normalizing by the $l_2$ norm plus $\xi$, and projecting with $\phi_r$. By Lemma A.2.2, each displacement rotates by $R$, hence its Euclidean norm is unchanged, implying that any scalar constructed purely from $||r_{i,c_i} - r_{j,c_j}||_2$ is rotation- and translation-invariant. The remaining operations $U_i, U_j, vec()$, normalization, and $\phi_r$ act in feature space and do not introduce dependence on the global coordinate frame. Therefore the final $g_{ij}$ is invariant under any $T \in E(3)$. This result is the key: FlexEGNN's geometric descriptor is a symmetry-respecting function of relative geometry only, hence does not leak global frame information.

## A.3. Flexibility Modulation as a Symmetry-Preserving Conditional Scalar Field

FlexEGNN introduces learnable modulation parameters $w_\Delta, b_\Delta$ that depend on the discretized sequence distance $\Delta_{i,j} = |p_i - p_j|$ and modulate edge aggregation as in Eq. (2).

**Lemma A.3.1 (Geometry-independence of $\mathbf{w_{\Delta_{i,j}}}, \mathbf{b_{\Delta_{i,j}}}$)**

For any rigid transformation $T \in E(3)$, the values $w_{\Delta_{i,j}}$ and $b_{\Delta_{i,j}}$ remain unchanged.

**Proof**

$\Delta_{i,j} = |p_i - p_j|$ is defined solely by residue indices and is independent of spatial coordinates. Hence applying $T$ cannot alter $\Delta_{i,j}$, and therefore cannot change $w_{\Delta_{i,j}}$ or $b_{\Delta_{i,j}}$.

**Corollary A.3.2 (Equivariance compatibility)**

Flexibility modulation changes the magnitude of edge contributions without introducing any global-frame dependent orientation bias.

**Justification**

Eq. (2) multiplies $\phi_e\left(h_i \,\|\, h_j \,\|\, g_{ij}\right)$ by a scalar $w_{\Delta_{i,j}}$ and shifts by a scalar $b_{\Delta_{i,j}}$. Since these scalars geometry-independent (Lemma A.3.1) and $g_{ij}$ is invariant (Proposition A.2.3), the modulation commutes with any rigid transformation and therefore preserves equivariance whenever the underlying coordinate update is constructed from relative displacements.

This establishes a crucial conceptual point: flexibility modulation is a sequence-conditioned scalar field defined on edges, not a geometric operator that could break E(3) symmetry.

## A.4. Equivariance of Channel-Weighted Centers and Coordinate Updates

FlexEGNN defines channel-weighted centers as in Eq. (4):

$$z_i = \sum_{c=1}^{C} \frac{w_{i,c}}{\sum_{c'=1}^{C} w_{i,c'}} r_{i,c}, \ \bar{z}_j = \sum_{c=1}^{C} \frac{w_{j,c}}{\sum_{c'=1}^{C} w_{j,c'}} r_{j,c} \tag{20}$$

**Lemma A.4.1 (E(3)-equivariance of centers)**

For any $T(r) = Rr + t$, the centers transform equivariantly:

$$z_i \to Rz_i + t, \ \bar{z}_j \to R\bar{z}_j + t \tag{21}$$

**Proof**

Each $z_i$ is a convex (normalized) linear combination of the channel coordinates $r_{i,c}$. Applying $T$ to each $r_{i,c}$ yields a convex combination of $Rr_{i,c} + t$, which equals $Rz_i + t$ because the weights sum to one. The same holds for $\bar{z}_j$.

FlexEGNN updates latent coordinates as in Eq. (5), where the update is expressed through relative terms of the form $(z_i - \bar{z}_j)$.

**Proposition A.4.2 (E(3)-equivariance of FlexEGNN coordinate update)**

Assume $\widetilde{\alpha}_{i,j}$ and all coefficients used in Eq. (5) are E(3)-invariant scalars (which holds because they are produced from node features and invariant descriptors such as $g_{ij}$). Then the coordinate update in Eq. (5) is E(3)-equivariant.

**Proof**

By Lemma A.4.1, $z_i - \bar{z}_j$ transforms as

$$(z_i - \bar{z}_j) \to (Rz_i + t) - (R\bar{z}_j + t) = R(z_i - \bar{z}_j) \tag{22}$$

i.e., as a 3D vector under rotation and is translation-invariant. Multiplying this vector by any invariant scalar $\widetilde{\alpha}_{i,j}$ preserves the vector transformation rule. Summation over neighbors and averaging do not alter equivariance. Finally, adding the equivariant base point $z_i \to (Rz_i + t)$ yields $\hat{z}_i \to (R\hat{z}_i + t)$. Therefore Eq. (5) is E(3)-equivariant.

**Corollary B.4.3 (Equivariance under flexibility modulation)**

Since flexibility modulation only rescales invariant message terms (Lemma A.3.1) and does not alter the relative-displacement structure of Eq. (5), FlexEGNN remains strictly E(3)-equivariant with flexibility modulation enabled.

### A.5. Theoretical Interpretation: Flexibility as Symmetry-Preserving Anisotropy

The above results imply that FlexEGNN implements a symmetry-preserving inductive bias. The flexibility modulation $\left(w_{\Delta_{i,j}}, b_{\Delta_{i,j}}\right)$ introduces edge-dependent interaction anisotropy controlled by sequence distance, enabling the same geometric pattern to exert different influence across distinct flexibility regimes. Importantly, this anisotropy is imposed through coordinate-free scalars and invariant geometric descriptors, hence it does not compromise E(3)-equivariance. Consequently, FlexEGNN is not merely "an EGNN with extra gates", but an explicitly constructed equivariant architecture whose expressive power is enhanced by a sequence-topology conditioned modulation field.

## B. Loss for Training the Flex-VAE

During training, for the amino acid sequence, a cross-entropy loss is computed on the set of masked residues $\Omega_S$ in $\widetilde{G_{aa}^P}$:

$$\mathcal{L}_{seq} = -\frac{1}{|\Omega_S|} \sum_{i \in \Omega_S} log p_D \left(S_i^{true} | H_i, Z_i, G_{aa}^B\right) \tag{23}$$

where $S_i^{true}$ denotes the ground-truth amino acid residue type, and $p_D$ is the sequence probability distribution predicted by the decoder. The Kullback-Leibler divergence of the structural latent variables is computed with respect to a standard isotropic Gaussian prior:

$$\mathcal{L}_{KL}^h = -\frac{1}{|G_{aa}^P|} \sum_{i \in G_{aa}^P} KL \left(q_\psi \left(H_i\right) | \mathcal{N} \left(0, I\right)\right) \tag{24}$$

where $q_\psi \left(H_i\right)$ denotes the Gaussian posterior of $H_i$ defined in Equation 8. For amino acid structures, the mean squared error of full-atom coordinates is first computed over the set of all valid atoms $\Omega_{atom}$, and an additional constraint is imposed on the backbone by computing the loss over the $C_\alpha$ atom set $\Omega_{CA}$:

$$\mathcal{L}_{coord}^{atom} = \frac{1}{|\Omega_{atom}|} \sum_{i \in \Omega_{atom}} ||\hat{H}_i - H_i||_2^2 \tag{25}$$

$$\mathcal{L}_{coord}^{CA} = \frac{1}{|\Omega_{CA}|} \sum_{i \in \Omega_{CA}} ||\hat{H}_i - H_i||_2^2 \tag{26}$$

The structural reconstruction loss is then obtained by combining these terms:

$$\mathcal{L}_{str} = \lambda_{atom}\mathcal{L}_{coord}^{atom} + \lambda_{CA}\mathcal{L}_{coord}^{CA} \tag{27}$$

where $\lambda_{atom}$ and $\lambda_{CA}$ are weighting hyperparameters. For the geometric latent variables, a zero-mean isotropic Gaussian prior is applied to the relative offsets $\Delta_{z_i}$, and the corresponding KL divergence term is given by:

$$\mathcal{L}_{KL}^z = -\frac{1}{|G_{aa}^P|} \sum_{i \in G_{aa}^P} KL \left(q_\psi \left(\Delta_{z_i}\right) | \mathcal{N} \left(0, I\right)\right) \tag{28}$$

where $q_\psi \left(\Delta_{z_i}\right)$ denotes the Gaussian posterior of $\Delta_{z_i}$ defined in Equation 9.

## C. The details of $E_{phys}$ and $\lambda(t)$ in TDEG

In our implementation, $E_{phys}$ acts neither as a black-box energy function derived from an external scorer nor as a test-time optimization objective directly obtained by invoking PyRosetta. Instead, it serves as a differentiable constraint term formulated from geometric and physical priors, defined as:

$$E_{phys} = E_{consec} + E_{inner} + E_{outer} + E_{angle} + E_{torsion} + E_{radius} \tag{29}$$

The term $E_{consec}$ represents the adjacent residue distance constraint, defined as:

$$E_{consec} = \sum_i max \left(0, d_i - u\right) + max \left(0, l - d_i\right) \tag{30}$$

where $d_i = ||z_{i+1} - z_i||$, the interval $[l, u] = [\mu - k\sigma, \mu + k\sigma]$, where $\mu$ and $\sigma$ denote statistics derived from the training data, and $k$ represents a tolerance coefficient.

The intra-peptide clash constraint $E_{inner}$ and the receptor clash constraint $E_{outer}$ are defined as:

$$E_{inner} = \sum_{|i-j|>1} max\left(0, \mu - ||z_i - z_j||\right) \tag{31}$$

$$E_{outer} = \sum_{i,j} max\left(0, \mu - ||z_i - x_j||\right) \tag{32}$$

The bond angle constraint $E_{angle}$ is defined as:

$$E_{angle} = \sum_{i} (\theta_i - \theta_0)^2 \tag{33}$$

where $\theta_0$ serves as a reference angle obtained from data statistics. The dihedral angle constraint $E_{torsion}$ is defined as:

$$E_{torsion} = \mathbb{E}\left[max\left(0, |\phi_i| - \phi_0\right)\right] \tag{34}$$

where $\phi_0$ originates from the statistical distribution properties of protein backbone dihedral angles characterized by the Ramachandran plot(Zhou et al., 2011). The spatial distribution constraint $E_{radius}$ is defined as:

$$E_{radius} = (std(||z_i - \bar{z}||) - r_0)^2 \tag{35}$$

where $r_0$ denotes a scale parameter derived from data statistics.

These components do not attempt to reconstruct a comprehensive molecular force field. Rather, they establish a lightweight and differentiable geometric energy suitable for guidance during the sampling phase, drawing upon physically inspired constraints commonly utilized in protein and peptide structure modeling, such as local geometric continuity, spatial repulsion, angular smoothness, and overall distribution stability.

In PepFGLD, the diffusion step $t$ is first normalized to $\tau = \frac{t}{T}$, where $T$ represents the total number of diffusion steps. The schedule $\lambda(t)$ is defined as:

$$\lambda(t) = \begin{cases} \lambda_{\min}, & \tau < \tau_0, \\ \lambda_{\min} + (\lambda_{\max} - \lambda_{\min}) \cdot \dfrac{\tau - \tau_0}{1 - \tau_0}, & \tau \geq \tau_0. \end{cases} \tag{36}$$

where $\lambda_{min}$ and $\lambda_{max}$ denote the lower and upper bounds of the guidance strength respectively, and $\tau_0$ marks the transition point from the exploration phase to the refinement phase. These variables serve as hyperparameters. We set $\lambda_{min}$ to 0.1, $\lambda_{max}$ to 0.7, and $\tau_0$ to 0.7. This parameter configuration maintains weak guidance during most of the denoising process to preserve exploration capacity and progressively strengthens the physical constraints only in the final stages to improve structural feasibility.

## D. Theoretical Analysis of TDEG-Guided Geometric Latent Diffusion

### D.1. Forward Diffusion and the Reverse Parameterization

In the geometric latent diffusion model, forward diffusion perturbs both $H_i$ and $Z_i$ via via Eq. (11). The reverse process is parameterized by a denoising network, and the sampling update is augmented by TDEG through Eq. (13). A central concern is whether injecting $\nabla_{Z_i} E_{phys}(Z)$ breaks diffusion correctness. We address this by placing Eq. (13) into a principled probabilistic framework: energy-tilted score guidance.

### D.2. Energy Guidance as Sampling from a Time-Dependent Tilted Distribution

Define the (implicit) marginal distribution of latent coordinates at timestep $t$ under the learned diffusion model as $p_t(Z)$. The denoiser $\epsilon_\theta^z$ is trained by the standard noise prediction objective Eq. (13), meaning it aims to approximate the diffusion score information needed to reverse the forward perturbation. TDEG adds a deterministic drift term proportional to $\nabla_z E_{phys}(Z)$ with a schedule $\lambda(t)$.

**Definition C.2.1 (Energy-tilted distribution)**

For each timestep $t$, define a tilted distribution

$$\hat{p}_t(Z) \propto p_t(Z)exp(-\lambda(t)E_{phys}(Z)) \tag{37}$$

**Proposition C.2.2 (TDEG corresponds to tilted-score guidance)**

Under the standard score-based interpretation of diffusion sampling, augmenting the reverse update by $\lambda(t)\nabla_z E_{phys}(Z)$ implements sampling biased toward $\hat{p}_t(Z)$, i.e., it modifies the effective score from $\nabla_z log p_t(Z)$ to $\nabla_z log \hat{p}_t(Z)$.

**Proof**

By Definition C.2.1,

$$log\hat{p}_t(Z) = log p_t(Z) - \lambda(t)E_{phys}(Z) + const \tag{38}$$

where, const $= -\log \int p_t(Z) \exp(-\lambda(t)E_{\text{phys}}(Z)) \, dZ$ denotes a normalization term independent of $Z$, which vanishes under gradient operations. Hence,

$$\nabla_z log\hat{p}_t(Z) = \nabla_z log p_t(Z) - \lambda(t)\nabla_z E_{phys}(Z) \tag{39}$$

Therefore, adding $\lambda(t)\nabla_z E_{phys}(Z)$ to the reverse drift is equivalent (up to sign conventions used by the chosen sampler discretization) to replacing the original score term with the score of the tilted distribution. This does not invalidate the probabilistic interpretation; it changes the target distribution in a controlled, explicitly stated manner. This result formalizes TDEG: it is not an ad-hoc force injection, but a principled time-dependent energy tilting of the generative distribution in latent space.

**D.3. Why the Time Schedule $\lambda(t)$ Matters**

Because $\lambda(t)$ is monotonically increasing and is designed to be small at high-noise timesteps, the sampling remains close to the original model distribution $p_t(Z)$ during the exploratory phase, while gradually emphasizing feasibility as noise decreases.

**Proposition C.3.1 (Asymptotic consistency at high-noise stages)**

If $\lambda(t) \to 0$ at high-noise timesteps, then $\hat{p}_t(Z)$ approaches $p_t(Z)$ in the sense that

$$\hat{p}_t(Z) \propto p_t(Z)(1 + o(1)) \tag{40}$$

**Proof**

When $\lambda(t) \to 0$, $\exp(-\lambda(t)E_{phys}(Z)) = 1 - \lambda(t)E_{phys}(Z) + o(\lambda(t))$, yielding $\hat{p}_t(Z) \propto p_t(Z)(1 + o(1))$ after normalization. Thus, TDEG preserves diversity at early steps and only introduces a physical bias near convergence, aligning with the intended design.

**D.4. Symmetry and Equivariance of $E_{phys}(Z)$ and Its Gradient**

A second concern is whether $E_{phys}(Z)$ breaks geometric symmetry or equivariance. The energy is described as a sum of terms involving consecutive residue distances, non-bonded collisions, bond angles and dihedrals, and spatial distribution. These quantities are standard rigid-motion invariants: they depend only on relative distances and angles.

**Assumption C.4.1 (Rigid-motion invariance of $E_{phys}$)**

For any rigid transformation $T \in E(3)$, the energy satisfies

$$E_{phys}(T(Z)) = E_{phys}(Z) \tag{41}$$

where $T(Z)$ denotes applying T to all latent coordinates. This assumption is satisfied when each component of $E_{phys}$ is expressed purely in terms of distances, angles, and dihedrals of latent points, because such quantities are invariant to rotation and translation.

**Proposition C.4.2 (Equivariance of the energy gradient)**

Under Assumption C.4.1, the gradient of the energy is E(3)-equivariant:

$$\nabla_{T(Z)} E_{phys}(T(Z)) = R\nabla_z E_{phys}(Z) \tag{42}$$

**Proof**

Consider $f(Z) = E_{phys}(Z)$ with $f(T(Z)) = f(Z)$. Differentiating both sides with respect to $Z$ using the chain rule yields

$$\nabla_z f(T(Z)) = (DT(Z))^T \nabla_{T(Z)} f(T(Z)) \tag{43}$$

where $DT(Z)$ is the Jacobian of $T$. For rigid transformations $T(Z) = RZ + t$, the Jacobian equals $R$. Therefore

$$\nabla_z f(T(Z)) = R^T \nabla_{T(Z)} f(T(Z)) \tag{44}$$

Because $f(T(Z)) = f(Z)$, we have $\nabla_z f(T(Z)) = \nabla_z f(Z)$, hence

$$\nabla_z f(Z) = R^T \nabla_{T(Z)} f(T(Z)) \rightarrow \nabla_{T(Z)} f(T(Z)) = R\nabla_z f(Z) \tag{45}$$

Substituting back $f = E_{phys}$ gives the claim.

**Corollary C.4.3 (Symmetry preservation of TDEG)**

Since $\nabla_z E_{phys}(Z)$ is equivariant, injecting $\lambda(t)\nabla_z E_{phys}(Z)$ into Eq. (13) preserves the E(3)-equivariance of the reverse update in $Z$-space.

The above analysis clarifies the exact theoretical status of TDEG. The forward diffusion and training objective remain unchanged; TDEG modifies the sampling-time reverse dynamics to target a time-dependent, energy-tilted distribution $\hat{p}_t(Z)$ while preserving rigid-motion symmetry, provided $E_{phys}$ is rigid-motion invariant. Therefore, the method does not break diffusion correctness; it performs a principled, schedule-controlled biasing of the generative distribution toward physically plausible latent geometries.

# E. The details of datasets

PepBench consists of protein-peptide complexes from the Protein Data Bank(Berman et al., 2000) that satisfy peptide length (4-25 residues) and receptor size requirements. After removing redundant entries with more than 90% sequence identity between receptors and peptides, a large supervised dataset is obtained. To enable cross-target generalization evaluation, the LNR dataset(Tsaban et al., 2022) is used as a test subset of PepBench. All PDB complexes are merged with the LNR set and clustered at a 40% receptor sequence similarity threshold using MMseqs2(Steinegger & Söding, 2017). Complexes belonging to the same clusters as LNR are removed from the training and validation sets to ensure that test receptor clusters are completely unseen during training. The remaining complexes are then randomly split at the cluster level to form the training and validation sets of PepBench. PepBDB is derived from protein-peptide binding complexes(Wen et al., 2019), and similar clustering-based splitting is applied to ensure non-redundant test sets, providing complementary evaluation and comparison. In addition, the ProtFrag dataset is constructed by fragmenting single-chain proteins from the PDB into stable segments of length 4-25 residues, and is used for unsupervised training.

To clarify, the PepGLAD results reported in Table 1 do not directly originate from the values in the original paper but come from retraining and evaluating the model under an experimental setup identical to that of PepFGLD. We attribute this discrepancy primarily to differences in data processing strategies. The original pipeline processes certain samples more leniently, such as ignoring missing local atoms, allowing them to remain in the training and evaluation phases. In contrast, we apply a stricter and uniform processing standard, retaining only the samples that stably pass through the entire pipeline. We then retrain and evaluate the baselines on this filtered data. To verify that this process introduces no distribution shift, we compare the data distributions before and after cleaning based on peptide length and calculate a Jensen-Shannon divergence of 0.0071, which indicates high distributional consistency.

## F. The details of baselines

- HSRN, which generates antibody sequences in a hierarchical autoregressive manner and progressively refines structure;

- dyMEAN, which employs a full-atom geometric encoder for antibody structure generation;

- GeoLDM, which first applies latent diffusion to the field of small-molecule generation;

- PepGLAD, which uses a variational autoencoder to encode variable-size full-atom residues into a fixed latent space and performs diffusion, while incorporating receptor-specific affine transformations for peptide design;

- PepMimic, which guides peptide design by mimicking the binding interface between the target and a known binding protein.

## G. The details of metrics

For evaluation metrics, in the Sequence-Structure Co-Design task, binding energy ($\Delta G$) based on the Rosetta energy function is used to reflect the physical feasibility and binding quality of the generated structures, measuring the interaction strength between candidate peptides and the receptor. The proportion of designs that satisfy both geometric and energy constraints is reported as the success rate (Success), indicating the overall physical plausibility of peptide structures. To characterize the exploratory capability of the model at the distribution level, Diversity (Div) is defined to measure exploration in both sequence space and conformational space, while Consistency (Con) is defined to quantify the degree of coupling between sequence variations and conformational changes. In the Binding Conformation Generation task, the root-mean-square deviation of $C_\alpha$ atoms $RMSD_\alpha$ is used to evaluate global backbone conformational error, the RMSD over all atomic coordinates $RMSD_a$ is used to assess fine-grained geometric quality, and the DockQ score is reported to measure atom-level interface agreement.

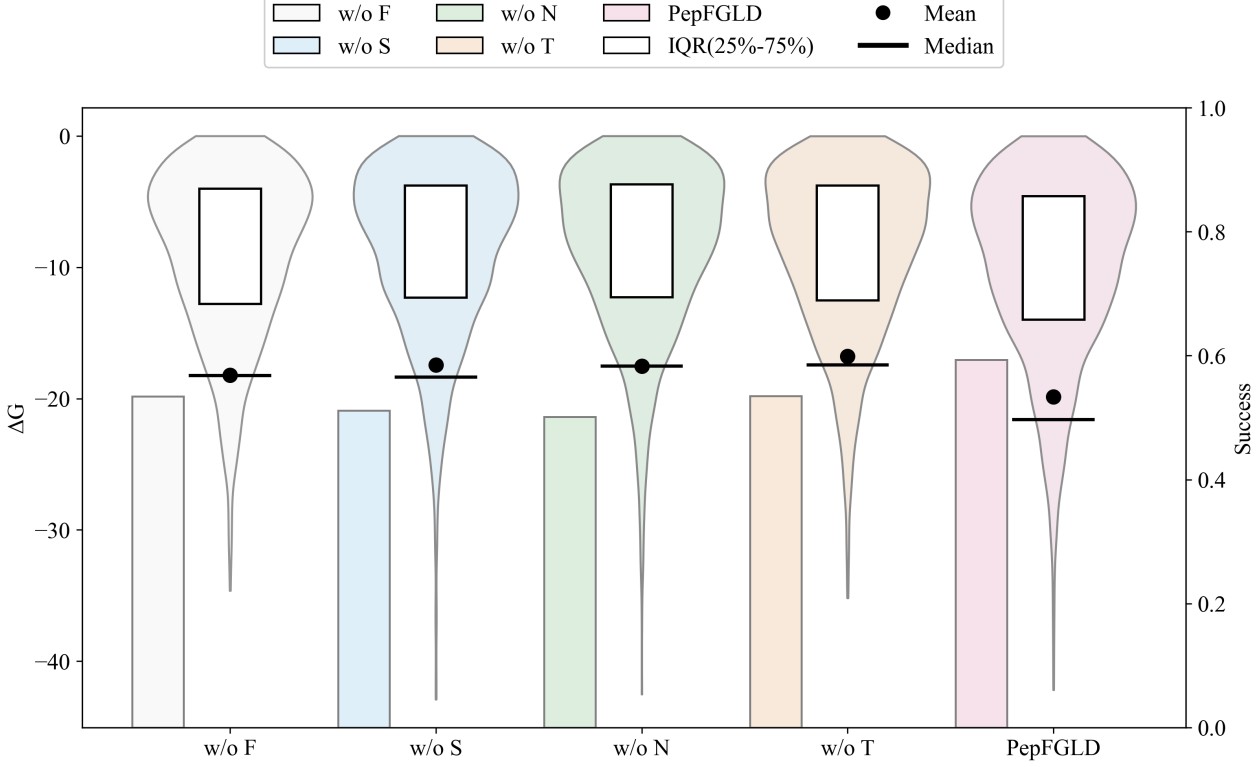

*Figure 5.* Ablation analysis of PepFGLD. Success rate is shown as bar plots, and the $\Delta G$ distributions are shown as violin plots.

*Table 6.* Ablation study of time-dependent energy guidance.

| Model | $\Delta G \downarrow$ | Success ↑ | Div ↑ | Con ↑ |
|---|---|---|---|---|
| w/o T | -16.77 | 53.52% | **0.6623** | 0.8323 |
| Constant $\lambda$ | -19.15 | 55.51% | 0.6193 | **0.8325** |
| PepFGLD | **-19.86** | **59.35%** | 0.5824 | 0.8160 |

## H. Ablation Analysis

To systematically evaluate the contributions of the core components in PepFGLD, four ablation variants are constructed: w/o F, in which FlexEGNN is removed and replaced with a standard EGNN; w/o S, in which SSBIM is removed; w/o N, in which the kernel-driven nonlinear latent variable mapping is removed and replaced with a linear mapping; and w/o T, in which TDEG is removed. For each variant, the same number of candidate structures is sampled per example. The $\Delta G$ values of all successful candidates are collected and visualized using violin plots to illustrate their distributions, while the Mean, Median, and IQR range (25%-75%) are reported to characterize the central tendency and dispersion of the energy distributions. The Success rate is used as an indicator of overall generation feasibility and is presented using bar plots. As shown in Fig. 5, all four key modules make substantive contributions to the final performance, although their impacts differ in magnitude. Removing FlexEGNN (w/o F) or SSBIM (w/o S) leads to a clear upward shift in the overall $\Delta G$ distribution and a substantial drop in success, indicating that flexibility-aware encoding and sequence-structure interaction are the primary factors enabling the generation of physically feasible conformations. Without these modules, the model becomes less responsive to local geometric variations, making generated conformations harder to adapt to receptor interfaces, which in turn increases energy scores and reduces success rates. In contrast, the w/o N exhibits a heavier tail in the $\Delta G$ distribution, suggesting that the kernel-driven nonlinear latent mapping plays an important role in stabilizing latent representations and reducing outlier conformations. The w/o T yields the worst mean and median $\Delta G$ among all ablations, with the entire energy distribution shifted toward higher-energy regions, and also shows a slightly lower success rate than the full model. This indicates that time-scheduled physical guidance not only affects whether generated samples satisfy physical constraints, but also critically determines the energy quality of successful samples.

In addition, we further considered the case with a constant $\lambda$. We include a control experiment with a constant $\lambda$=0.5, and the table presents the results. Experimental results indicate that, compared to the w/o T setting, energy guidance with a constant $\lambda$ improves performance. Specifically, $\Delta G$ improves from -16.77 to -19.15 and Success increases from 53.52% to 55.51%, which confirms the effectiveness of the energy constraint itself. Compared to the constant $\lambda$ configuration, PepFGLD achieves further enhancements in both $\Delta G$ and Success, validating the effectiveness of $\lambda(t)$. The results also reveal that the Con and Div metrics of PepFGLD experience a slight decline compared to the two variants. This occurs because the time schedule progressively strengthens the constraint in the later stages, which concentrates the generation distribution more tightly on physically feasible regions. Consequently, this mechanism partially compresses the exploration space but yields superior binding quality and overall feasibility.

## I. Supplementary information in Chapter 5.4

To further assess the stability and applicability of PepFGLD under different peptide characteristics and sampling conditions, three groups of extended experiments are conducted on PepBench for the Sequence-Structure Co-Design task. Generation quality is comprehensively evaluated using $\Delta G$, mean full-atom, $C_\alpha RMSD$, and DockQ. PepGLAD and PepMimic, which show the strongest performance among the baselines, are selected for comparison. Figs. 3(a) and 6(a, d) illustrate model performance across three peptide length regimes: short peptides (4-8 residues), medium-length peptides (9-15 residues), and long peptides (16-25 residues). The $\Delta G$ distributions show that PepFGLD consistently maintains lower energy levels across all three length ranges, with particularly stable performance in the medium and long regimes, indicating a stronger ability to adapt to larger conformational spaces. Trends in RMSD and DockQ further demonstrate that PepFGLD achieves better geometric recovery than the baselines across different peptide scenarios. PepGLAD shows some advantage in the short-peptide regime, while the performance of PepMimic exhibits stronger dependence on peptide length. Figs. 3(b) and 6(b, e) compare model performance across three secondary structure types: helix, strand, and loop. PepFGLD achieves lower $\Delta G$ and higher DockQ for strand and loop structures, indicating an advantage in handling regions with larger conformational variability. In contrast, performance differences among the models become more limited in the more rigid helix regime. The RMSD results further reflect this trend: PepFGLD maintains lower errors for loop structures, whereas the baselines exhibit

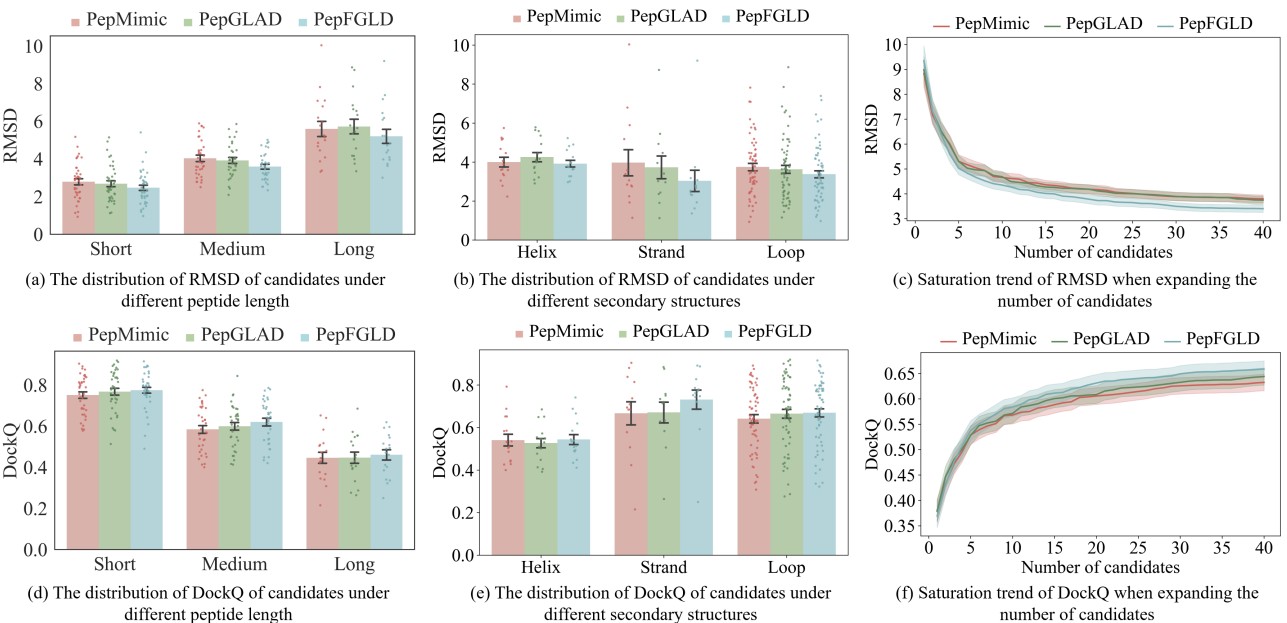

*Figure 6.* Evaluation of models under different peptide characteristics on PepBench.

greater dispersion in this regime, suggesting persistent instability when modeling flexibility-dominated conformational changes. Figs. 3(c) and 6(c, f) illustrate the saturation behavior as the number of sampled candidates increases from 1 to 40. All three metrics exhibit typical convergence patterns, but PepFGLD shows a faster rate of improvement: $\Delta G$ reaches the low-energy regime earlier, and both RMSD and DockQ stabilize with fewer samples. This indicates that, under the same sampling budget, PepFGLD is more likely to generate high-quality candidate structures and can achieve optimal performance without requiring a large number of samples.

## J. Supplementary information in Chapter 5.5

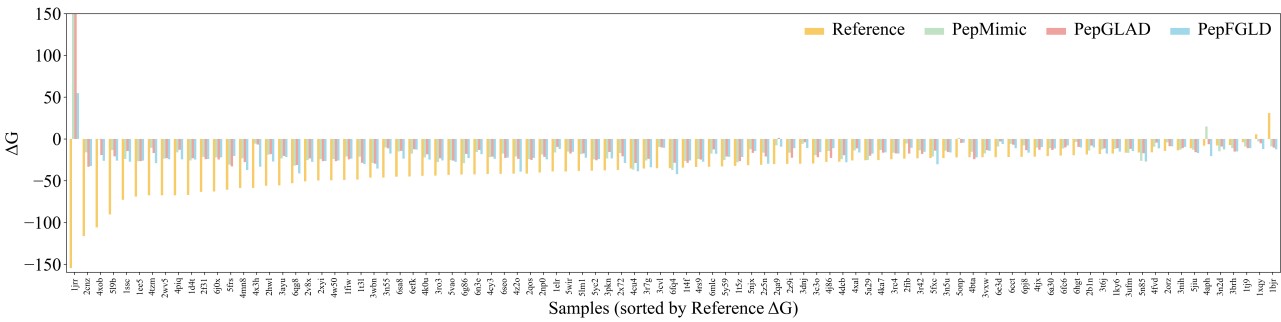

*Figure 7.* Comparison of $\Delta G$ distributions between references and best candidates.

This analysis aims to assess whether the model can identify low-energy conformations that are comparable to or better than the reference in energy space, whether the overall energy distribution of generated structures aligns with the reference distribution, and whether consistent trends are observed as the difficulty of reference samples varies. To quantitatively characterize the discrepancy between the energy distribution of generated conformations and that of the reference structures, several metrics are designed and used to compare the three methods. Fig. 7 presents a per-sample comparison between each model and the reference $\Delta G$ distribution. All three methods are able to find conformations with lower energy than the reference for a subset of samples; however, clear differences in distributional behavior are observed. PepMimic exhibits larger deviations with more high-energy outliers, PepGLAD shows a moderate level of consistency with the reference trend, whereas PepFGLD remains closely aligned with the reference range for most samples and substantially reduces high-energy

deviations.

To evaluate the performance of different models in predicting the best candidate ΔG on PepBench from ranking consistency, energy deviation magnitude, and overall statistical difference, we adopt Spearman correlation, RMSE, and paired t-test statistics as evaluation metrics. Assume that the dataset contains $N$ samples. For the $i$-th sample, $\Delta G_i^r$ denotes the reference energy value, and $\Delta G_i^m$ denotes the lowest energy among all candidate conformations generated by the model.

Spearman-$\rho$ is used to quantify the rank consistency between the predicted energies and the reference energies across samples. It is defined as:

$$\rho = 1 - \frac{6 \sum_{i=1}^{N} (R_i^m - R_i^r)^2}{N(N^2 - 1)} \tag{46}$$

where $R_i^m$ and $R_i^r$ represent the ranks of $\Delta G_i^m$ and $\Delta G_i^r$ among all samples, respectively.

p(Spearman) denotes the two-sided p-value associated with Spearman-$\rho$, which measures the probability of observing the corresponding rank correlation under the null hypothesis of no association.

Root Mean Square Error (RMSE) measures the overall magnitude of deviation between the predicted energies and the reference energies and is defined as:

$$\text{RMSE} = \sqrt{\frac{1}{N} \sum_{i=1}^{N} (\Delta G_i^m - \Delta G_i^r)^2} \tag{47}$$

Paired t-statistic (t-stat) is used to examine whether there exists a systematic shift between the predicted energies and the reference energies. Let

$$t = \frac{\bar{d}}{s_d / \sqrt{N}} \tag{48}$$

where $\bar{d}$ denotes the sample mean of $\{\Delta G_i^m - \Delta G_i^r\}$ and $s_d$ denotes the corresponding sample standard deviation.

p(t) represents the two-sided p-value of the paired t-test, which quantifies the statistical significance of the observed energy difference.

## K. Case Studies of Peptide Generation under Structural Constraints

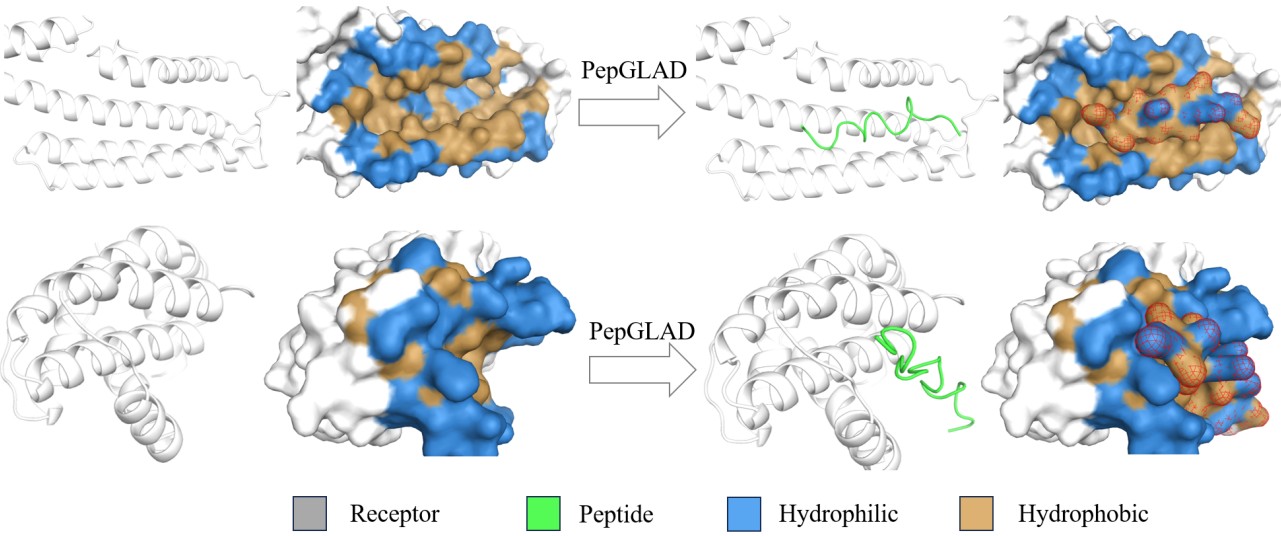

*Figure 8.* Top: A generated candidate confined within the binding site (PDB=6fq4, ΔG=-28.54). Bottom: A generated candidate with complementary shape to the binding site (PDB=6qg8, ΔG=-31.18). Both candidates form compact interactions at the interface.

To further evaluate peptide generation under receptor-imposed structural constraints, we select two representative protein–peptide complexes (PDB: 6fq4 and 6qg8), each presenting a distinct conformational challenge.

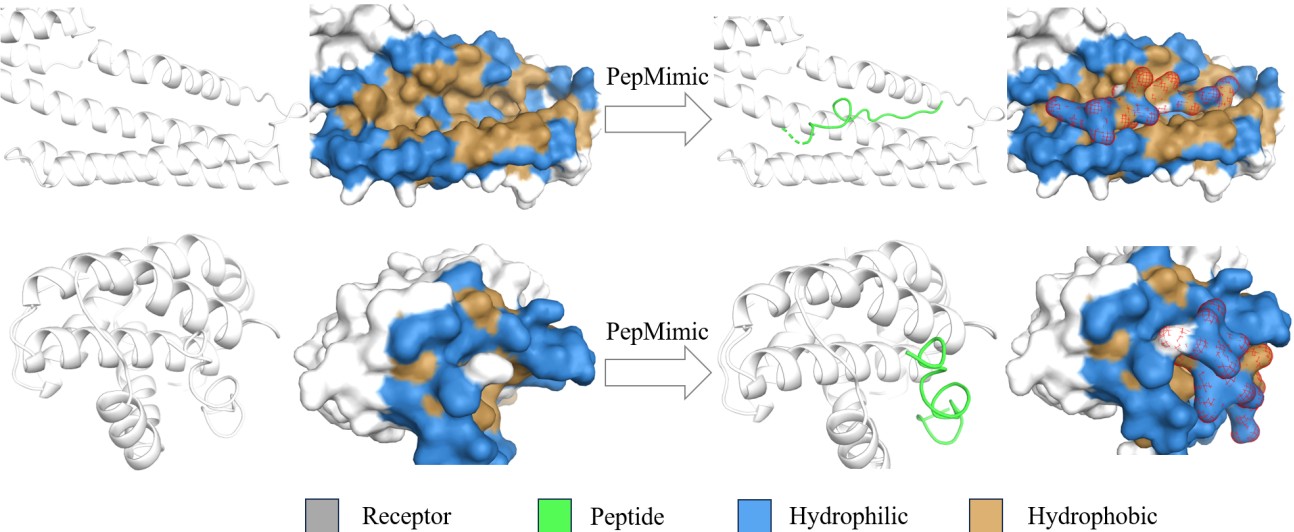

*Figure 9.* Top: A generated candidate confined within the binding site (PDB=6fq4, ΔG=-37.13). Bottom: A generated candidate with complementary shape to the binding site (PDB=6qg8, ΔG=-32.04). Both candidates form compact interactions at the interface.

In 6fq4, the endogenous peptide follows a relatively smooth trajectory along the receptor surface. Its accessible conformational space is narrowly restricted by the binding groove, imposing strict requirements on local geometric accuracy, directional continuity, and interface complementarity. In contrast, 6qg8 features multiple folded segments with pronounced directional changes. The interface is more open and the peptide exhibits high conformational freedom, making the generation task heavily dependent on the model's ability to explore global folding pathways in a high-dimensional energy landscape. Specifically, we generate 40 candidate peptides for each receptor using PepGLAD, PepMimic, and PepFGLD, and analyze the candidate with the lowest predicted binding free energy ΔG (Figs. 4 A and 8-9).

In the constrained geometry of 6fq4, PepGLAD and PepMimic both produce plausible conformations confined within the groove, forming reasonable interface contacts, with ΔG values of -28.54 kcal/mol and -37.13 kcal/mol, respectively. PepFGLD, however, yields a peptide with more coherent residue orientations, improved groove complementarity, and smoother backbone continuity, achieving a significantly lower ΔG of -42.18 kcal/mol, indicating superior modeling of local flexibility under strong geometric restrictions. Meanwhile, In the more complex 6qg8 system, all three methods generate peptides with broadly complementary shapes, but the performance differences become more pronounced. PepGLAD and PepMimic produce surface-fitting conformations with ΔG values of -31.18 kcal/mol and -32.04 kcal/mol, respectively. PepFGLD generates a peptide with a more plausible folding trajectory, better interface packing, and enhanced hydrophobic–hydrophilic matching, resulting in a substantially lower ΔG of -41.24 kcal/mol. These results indicate that when substantial conformational rearrangements are required, the flexibility-aware diffusion mechanism of PepFGLD enables more effective structural exploration and more stable energetic convergence.

Overall, across both geometrically constrained settings (e.g., 6fq4) and high-flexibility systems requiring extensive folding and spatial reorganization (e.g., 6qg8), PepFGLD consistently delivers better interface adaptation, more stable structural quality, and lower-energy peptide conformations, demonstrating its core advantage in flexible peptide modeling and physically coherent peptide generation.

To qualitatively assess generative performance, we illustrate a peptide conformation generated for a representative PepBench complex (PDB: 4KA7), which closely matches the experimental reference structure. At the same time, the characteristic structural environment of 4KA7 provides important context for this comparison. The endogenous peptide binds within a shallow and elongated surface groove that demands continuous backbone extension, where even minor angular deviations can disrupt the overall binding geometry. Owing to its compact local architecture and restricted backbone flexibility, 4KA7 is highly sensitive to subtle perturbations and therefore serves as an ideal test system for evaluating backbone stability in generative models.

As shown in Fig. 4 B, PepFGLD accurately reconstructs a continuous backbone that closely follows the native trajectory, achieving low RMSD values and a high DockQ score. In contrast, PepGLAD and PepMimic yield fragmented peptide

conformations in which the backbone breaks into disconnected segments, despite partially recovering the overall binding orientation. Such discontinuities substantially diminish geometric plausibility and impair receptor–peptide complementarity.

Specifically, PepGLAD and PepMimic perform diffusion over residue-level latent embeddings that do not explicitly preserve covalent topology or encode local flexibility, making them susceptible to abrupt structural deviations in geometry-constrained environments such as the 4KA7 groove. PepFGLD, by contrast, integrates flexibility-aware geometric encoding, bidirectional sequence–structure interaction, and a smooth nonlinear latent manifold. FlexEGNN adaptively modulates geometric relationships according to local conformational states, SSBIM aligns sequence semantics with spatial variation, and the nonlinear latent mapping shapes a continuous conformational manifold that suppresses abrupt geometric transitions. Although no single component explicitly enforces chain continuity, their integration constrains the latent space to remain within geometrically compatible regions, making abrupt backbone discontinuities unlikely.

## L. Algorithm for Training and Sampling

We present the pseudo codes for training in Algorithm 1 amd sampling in Algorithm 2.

## M. Hyperparameter settings

Flex-VAE: embedding size: 128; layer: 3; $\rho$: 0.5; $\lambda_h$: 0.3; $\lambda_z$: 0.5; $\lambda_{atom}$: 0.5; $\lambda_{CA}$: 1.0.

LDM: embedding size: 128; layer: 3; steps: 100.

---

**Algorithm 1** Training Algorithm of PepFGLD

---

1: **input** channel-augmented complexes $\mathcal{C}$
2: **output** encoder $E_\phi$, decoder $D_\xi$, denoising network $\epsilon_\theta$
3: **function** TrainFlexVAE($\mathcal{C}$)
4: Initialize $E_\phi, D_\xi$
5: **while** $\phi$ and $\xi$ have not converged **do**
6:     Sample $(G_{aa}^P, G_{aa}^B) \sim \mathcal{C}$
7:     $\Omega_S \leftarrow$ RandomMaskResidues($G_{aa}^P$)                                       {Mask residues}
8:     $\widetilde{G_{aa}^P} \leftarrow$ RemoveSideChain($G_{aa}^P, \Omega_S$)                       {Geometry perturbation}
9:     $\{\hat{h}_i\} \leftarrow$ FlexEGNN($\widetilde{G_{aa}^P}, G_{aa}^B$)                           {Equivariant encoding}
10:     $(\hat{S}_i, \hat{\hat{h}}_i) \leftarrow$ SSBIM($S_i, \hat{h}_i$)                               {Seq-geom alignment}
11:     $(\mu_i^h, \log \sigma_i^h) \leftarrow \psi_H(\hat{\hat{h}}_i)$                                {Posterior for $H$}
12:     $H_i \leftarrow \mu_i^h + \exp(\frac{1}{2} \log \sigma_i^h) \odot \varepsilon_i^h, \; \varepsilon_i^h \sim \mathcal{N}(0, I)$            {Reparam.}
13:     $\sigma_i^z \leftarrow \exp(\frac{1}{2} \psi_Z(\hat{h}_i))$                              {Scale for $\Delta_z$}
14:     $\Delta_{z_i} \leftarrow \sigma_i^z \odot \varepsilon_i^z, \; \varepsilon_i^z \sim \mathcal{N}(0, I)$                   {Zero-mean posterior}
15:     $Z_i \leftarrow \bar{\bar{z}}_i + \hat{\Delta}_{z_i}$                                 {Anchor-based latent}
16:     $\widehat{G_{aa}^P} \leftarrow D_\xi((H_i, Z_i), G_{aa}^B)$                             {Decoding}
17:     $\mathcal{L}_{VAE} \leftarrow (1 - \rho)\mathcal{L}_{seq} + \rho\mathcal{L}_{str} + \lambda_h \mathcal{L}_{KL}^h + \lambda_z \mathcal{L}_{KL}^z$     {Objective}
18:     $\phi, \xi \leftarrow$ optimizer($\mathcal{L}_{VAE}; \phi, \xi$)
19: **end while**
20: **return** $E_\phi, D_\xi$
21: **end function**
22: **function** TrainLatentDiffusion($E_\phi, D_\xi, \mathcal{C}$)
23: Initialize $\epsilon_\theta$
24: **while** $\theta$ has not converged **do**
25:     Sample $(G_{aa}^P, G_{aa}^B) \sim \mathcal{C}$
26:     $(H_i, Z_i) \leftarrow E_\phi(G_{aa}^P, G_{aa}^B)$                              {Latent encoding}
27:     $t \sim$ Uniform($\{1, \ldots, T\}$)                                {Time sampling}
28:     $\varepsilon_i^h, \varepsilon_i^z \sim \mathcal{N}(0, I)$
29:     $H_i^t \leftarrow \sqrt{\alpha^t} H_i + \sqrt{1 - \alpha^t}\, \varepsilon_i^h$                        {Forward diffusion}
30:     $Z_i^t \leftarrow \sqrt{\alpha^t} Z_i + \sqrt{1 - \alpha^t}\, \varepsilon_i^z$                        {Forward diffusion}
31:     $(\hat{\varepsilon}_i^h, \hat{\varepsilon}_i^z) \leftarrow \epsilon_\theta(H^t, Z^t, G_{aa}^B, t)$                    {Noise prediction}
32:     $\mathcal{L}_{LDM} \leftarrow \sum_i (\|\varepsilon_i^h - \hat{\varepsilon}_i^h\|_2^2 + \|\varepsilon_i^z - \hat{\varepsilon}_i^z\|_2^2)/|G_{aa}^P|$
33:     $\theta \leftarrow$ optimizer($\mathcal{L}_{LDM}; \theta$)
34: **end while**
35: **return** $\epsilon_\theta$
36: **end function**
37: $E_\phi, D_\xi \leftarrow$ TrainFlexVAE($\mathcal{C}$)
38: Fix parameters $\phi$ and $\xi$                                                 {Two-stage training}
39: $\epsilon_\theta \leftarrow$ TrainLatentDiffusion($E_\phi, D_\xi, \mathcal{C}$)
40: **return** $E_\phi, D_\xi, \epsilon_\theta$

---

---

**Algorithm 2** Sampling Algorithm of PepFGLD

---

1: **input** decoder $D_\xi$, denoising network $\epsilon_\theta$, binding site $G_{aa}^B$
2: **output** generated peptide $G_{aa}^P$
3: $P \leftarrow$ absolute positional embedding of peptide residues
4: $H^T \sim \mathcal{N}(0, I), \ Z^T \sim \mathcal{N}(0, I)$         {Latent initialization}
5: **for** $t = T, T-1, \ldots, 1$ **do**
6:    $(\hat{\epsilon}^h, \hat{\epsilon}^z) \leftarrow \epsilon_\theta(H^t, Z^t, P, G_{aa}^B, t)$         {Latent denoising}
7:    $H^{t-1} \leftarrow \frac{1}{\sqrt{\alpha^t}}\left(H^t - \frac{\beta_t}{\sqrt{1-\alpha^t}}\hat{\epsilon}^h\right) + \beta_t\eta^h, \ \eta^h \sim \mathcal{N}(0, I)$         {Reverse update}
8:    $\hat{Z}^{t-1} \leftarrow \frac{1}{\sqrt{\alpha^t}}\left(Z^t - \frac{\beta_t}{\sqrt{1-\alpha^t}}\hat{\epsilon}^z\right) + \beta_t\eta^z, \ \eta^z \sim \mathcal{N}(0, I)$         {Reverse update}
9:    $Z^{t-1} \leftarrow \hat{Z}^{t-1} + \lambda(t)\nabla_{Z_i}E_{phys}(Z^{t-1})$         {TDEG guidance}
10: **end for**
11: $G_{aa}^P \leftarrow D_\xi((H^0, Z^0), G_{aa}^B)$         {Full-atom decoding}
12: **return** $G_{aa}^P$

---

