# OpenReview forum: "Flexibility-Aware Geometric Latent Diffusion for Full-Atom Peptide Design"
_ICML.cc/2026/Conference — ICML 2026 regular_

### Official Review · Reviewer_DJnB · 2026-03-04

**Soundness:** 2
**Presentation:** 2
**Significance:** 2
**Originality:** 3
**Overall Recommendation:** 3
**Confidence:** 4

**Summary:**

This paper proposes PepFGLD, a receptor-conditioned full-atom peptide design framework that explicitly targets peptide flexibility and the strong coupling between sequence and conformation. The method first learns a joint latent representation of peptide sequence and geometry via a Flex-VAE, then trains a latent diffusion model to generate latent point-cloud states, and finally applies a time-dependent energy guidance (TDEG) during sampling. The stated motivation of TDEG is to preserve exploration and diversity at high noise levels while progressively enforcing physical/geometric feasibility at low noise levels. The architecture further includes a flexibility-aware equivariant structural encoder (FlexEGNN) and a bidirectional sequence--structure interaction module (SSBIM) to couple sequence semantics with conformational dynamics. Experiments on PepBench/PepBDB evaluate both sequence--structure co-design and binding conformation generation using Rosetta-based scores, success rate, diversity/consistency, and RMSD, accompanied by ablations and qualitative examples.

**Compliance With Llm Reviewing Policy:**

Affirmed.

**Final Justification:**

My concerns have been adequately addressed.

**Key Questions For Authors:**

1. Please provide a fully reproducible definition of $E_{\mathrm{phys}}$, including explicit formulas for all terms, weights, thresholds/smoothing, unit normalization, and whether gradients are taken w.r.t. latent $Z$ or decoded all-atom coordinates; additionally, specify $\lambda(t)$ in closed form or as a schedule table with all hyperparameters. Clarifying this would directly affect my soundness assessment.

2. How do you rule out evaluator alignment or test-time optimization effects, given that the primary metrics are Rosetta-based and TDEG applies energy guidance at sampling time? Can you provide at least one orthogonal validation (alternative docking/scoring, different force field, or short MD stability) to confirm the main conclusions? A positive answer would substantially improve my assessment of soundness and significance.

3. Please clarify baseline adaptation details and explain why some baselines fail extremely (e.g., unusually large RMSD). Did you reproduce each baseline under its original setting or report a sanity check close to published performance? Are training budgets, inputs/outputs, and sampling settings matched to PepFGLD as closely as possible? This will affect my confidence in the comparative claims.

4.  To support the ``flexibility-aware'' claim, can you provide stratified evaluation on high-flexibility subsets (by peptide length, disorder proxies, or conformational diversity proxies) showing that relative gains increase with flexibility? Without this, it remains unclear whether flexibility modeling is the main contributor.

5. Please provide complete pretrain-to-finetune protocol details (what is frozen/unfrozen in each stage, steps/epochs, data ratios, noise schedules) and sensitivity analyses for key choices. This would strengthen reproducibility and contribution attribution.

**Limitations:**

\No. The limitations discussion is not sufficient. The paper should explicitly discuss (i) dependence on Rosetta-like evaluators and the risk of metric alignment when using sampling-time guidance, (ii) potential fairness issues from baseline adaptation and compute/data budget differences, and (iii) potential dual-use considerations for peptide design and any release constraints.

**Strengths And Weaknesses:**

The paper addresses an important and timely problem in peptide design, where flexibility and multimodality make receptor-conditioned generation challenging. The proposed pipeline is coherent end-to-end, combining latent generative modeling with equivariant geometric encoders and explicit sequence--structure interaction, and the experimental section includes ablations that attempt to justify the contribution of major components.

However, there are several issues that substantially limit my confidence in the central claims.
1. The strongest performance gains appear to rely heavily on sampling-time energy guidance (TDEG), yet the paper does not specify the physical/geometric energy function $E_{\mathrm{phys}}$ at a reproducible level. In particular, the exact terms, weights, thresholds/smoothing, unit normalization, and whether gradients are taken with respect to the latent coordinates $Z$ or decoded all-atom coordinates are not described precisely enough for independent replication. The explicit functional form and hyperparameters of the guidance schedule $\lambda(t)$ are also underspecified. Without these details, it is difficult to attribute improvements to the learned model rather than to undocumented test-time optimization choices.

2. The evaluation protocol raises a serious evaluator-alignment concern. The primary metrics rely heavily on Rosetta-related scoring, while the method explicitly injects ``physical/geometric'' energy gradients during sampling. Without orthogonal validation (e.g., an alternative docking/scoring function, a different force field, or even short MD stability checks), it is hard to rule out the possibility that the proposed guidance is implicitly aligned to the evaluator, i.e., improving the reported score without necessarily improving true physical plausibility or generalization. This concern is particularly salient when guidance is applied at test time and the main metric is energy-based.

3. Baseline fairness is unclear. Some baselines appear to fail extremely (e.g., implausibly large RMSD values), which often indicates adaptation mismatch, implementation issues, or budget misalignment rather than a meaningful scientific comparison. Because the method uses a pretrain-to-finetune pipeline, it is critical to report whether baselines receive comparable data and compute budgets, and whether baseline implementations are verified under their original settings. In the current form, these uncertainties weaken the credibility of the SOTA claims.

4. The paper's ``flexibility-aware'' thesis is not directly validated. Although the method is motivated by flexibility, the experiments do not provide targeted stratified analyses on high-flexibility regimes (e.g., longer peptides, disorder proxies, or conformational diversity proxies) to demonstrate that gains correlate with flexibility rather than reflecting generic regularization or energy-guidance effects. The training protocol also remains insufficiently transparent in key details (freezing/unfreezing, stage-wise steps/epochs, data ratios, noise schedules), further complicating reproducibility and attribution.

---

> ### Author Rebuttal · Authors · 2026-03-30
>
> We thank the reviewer allowing us to improve our work.
>
> **$E_{phys}$ and $\lambda\left(t\right)$.** In PepFGLD, $E_{phys}$ is defined as
> $E_{phys}=E_{consec}+E_{inner}+E_{outer}+E_{angle}+E_{torsion}+E_{radius}$
>
> The consecutive distance constraint is
> $E_{consec}=\sum_i \big[\max(0,d_i-u)+\max(0,l-d_i)\big]$,
> where $d_i=\|z_{i+1}-z_i\|$ and $[l,u]=[\mu_t-k\sigma_t,\mu_t+k\sigma_t]$ with statistics derived from the training data $(\mu_t,\sigma_t)$ and tolerance $k$.
>
> The clash constraints are
> $E_{inner}=\sum_{|i-j|>1}\max(0,\mu_t-\|z_i-z_j\|)$ and
> $E_{outer}=\sum_{i,j}\max(0,\mu-\|z_i-x_j\|)$.
>
> The geometric constraints are
> $E_{angle}=\sum_i(\theta_i-\theta_0)^2$ and
> $E_{torsion}=\mathbb{E}[\max(0,|\phi_i|-\phi_0)]$,
> where $\theta_0$ and $\phi_0$ are statistics (the latter from the Ramachandran distribution).
>
> The spatial constraint is
> $E_{radius}=(\mathrm{std}(\|z_i-\bar{z}\|)-r_0)^2$,
> where $r_0$ is a scale parameter.
>
> The diffusion step is normalized as $\tau=t/T$. The guidance schedule is
> $\lambda(t)=\lambda_{\min}$ if $\tau<\tau_0$, and
> $\lambda(t)=\lambda_{\min}+(\lambda_{\max}-\lambda_{\min})\frac{\tau-\tau_0}{1-\tau_0}$ otherwise,
> where $(\lambda_{\min},\lambda_{\max})$ are bounds and $\tau_0$ is the transition point. We use $\lambda_{\min}=0.1$, $\lambda_{\max}=0.7$, and $\tau_0=0.7$.
>
> **Evaluator alignment.** As previously noted, the $E_{phys}$ employed in TDEG and the PyRosetta energy function used for evaluation represent entirely different systems. We agree that an additional evaluation would strengthen the paper, we evaluate PepFGLD and PepGLAD using Prime MM-GBSA, a post hoc scoring method under an alternative physics-based framework, where lower scores indicate more favorable binding. We avoided re-docking because evaluating an already bound complex with additional pose-sampling could alter the generated conformations of the peptides, making the results dependent on the docking procedure rather than the complex itself. Instead, MM-GBSA directly evaluates the generated bound state using static post hoc scoring without short MD stability analysis. The results demonstrate that PepFGLD achieves a higher candidate success rate (0.8758 vs. 0.8508). Among the successful samples, the best candidate structures of PepFGLD yield a superior average binding energy (-102.48 vs. -99.73). Consistent with the conclusions drawn from PyRosetta, this alternative scoring system confirms that the performance improvements reflect genuine enhancements in structural quality and binding affinity, rather than an alignment with a specific evaluator.
>
> **Baseline adaptation.** For fair comparison, all baselines were retrained and evaluated in a unified environment using their public codes and configurations. Regarding anomalous baseline performance like large RMSD, analysis reveals this primarily occurs in GeoLDM's Binding Conformation Generation on the PepBench. Conversely, GeoLDM maintains stability across datasets in sequence-structure co-design and achieves the best RMSD on PepBDB. As originally stated, GeoLDM underperforms on the cross-target generalization dataset PepBench but remains competitive on the homologous PepBDB, which suggest that the phenomenon likely stems from differences in model adaptability within cross-target generalization scenarios rather than issues related to reproducibility.
>
> **Flexibility-aware.** To support our flexibility-aware claim, we proxy structural flexibility using the peptide coil (C) residue proportion, since coils lack regular hydrogen-bonding patterns and exhibit higher flexibility. Using the I-TASSER PSSpred module, we predict sequence secondary structures (helix H, beta-sheet E/B, coil C) and calculate coil fractions. This metric partitions the test set into H-Flex and L-Flex subsets to evaluate PepFGLD against baselines. Aligning with classical conformational entropy penalties, all methods yield worse ∆G on the H-Flex. However, PepFGLD achieves ∆G improvements of 7.43 and 10.65 over baselines here, significantly exceeding its 4.91 and 5.78 gains on the L-Flex.
>
> | Method|Subset|∆G$\downarrow$|Success$\uparrow$|
> |---------|----------|----------------------|--------------------|
> |PepFGLD| H-Flex |**-14.54**| **53.00%**|
> ||L-Flex|**-27.57**|**68.55%**|
> |PepGLAD| H-Flex |-7.11|43.18%|
> ||L-Flex|-22.66|52.83%|
> |PepMimic|H-Flex|-3.89|46.59%|
> ||L-Flex|-21.79| 60.53%|
>
> **Hyperparameters.**  We agree that a complete training protocol is essential for reproducibility and have clarified it in the revision. PepFGLD uses a two-stage training strategy: Flex-VAE is first trained to learn stable latent representations and then frozen; the LDM is subsequently trained in this space, with unsupervised pretraining on protein fragments followed by supervised fine-tuning on protein-peptide complexes.
>
> Flex-VAE: layer: 3; $\rho$: 0.5; $\lambda_h$: 0.3; $\lambda_z$: 0.5; $\lambda_{atom}$: 0.5; $\lambda_{CA}$: 1.0; epoch: 100.
> LDM: layer: 3; steps: 100; epoch: pretrain 100, fine-tune 500.

---

> > ### Author Rebuttal · Reviewer_DJnB · 2026-04-03
> >
> > My concerns have been adequately addressed.

---

> > > ### Author Response · Authors · 2026-04-04
> > >
> > > Thank you for your kind response, and we're glad to see that our responses have addressed your concerns.

---

### Official Review · Reviewer_MHVF · 2026-03-08

**Soundness:** 3
**Presentation:** 3
**Significance:** 3
**Originality:** 3
**Overall Recommendation:** 5
**Confidence:** 4

**Summary:**

The paper addresses the problem of peptide design given the target with latent diffusion models. The innovations are from multiple aspects, covering model architecture, training strategies, and sampling augmentations. First, the authors improve the backbone of multi-channel equivariant graph neural network (EGNN) with residue-pair gating mechanisms, modeling the flexibility in geometric interactions. Second, the authors introduce sequence-structure interaction module to encourage mutual mixing of sequence information and structural geometry. Third, the authors explore two-stage training strategies to pretrain on protein fragments and finetune on protein-peptide data. Fourth, the authors integrate energy guidance during diffusion to encourage better physical validity of the generated samples. Experiments on existing benchmarks for design and docking demonstrate performance gain of the proposed model over baselines with a large margin.

**Compliance With Llm Reviewing Policy:**

Affirmed.

**Final Justification:**

The rebuttal fully solved my concerns.

**Key Questions For Authors:**

1. In section 3, the geometry of each residue is represented as three channels. What is the detailed definitions of the three channels?
2. How is the success rate of binding energy defined? Is a design considered successful if it has a $\Delta G<0$?

**Limitations:**

The limitation could be the computational efficiency problem induced by the requirements of training multiple modules, as well as multiple stages.

**Strengths And Weaknesses:**

**Strengths**

1. The authors include multiple innovations covering different aspects of the system, including model backbone, training strategy, and diffusion sampling, which contribute considerable knowledge to the community.
2. The evaluations are extensive and solid, exhibiting large improvement of the proposed model over baselines. Ablation studies are also informative for analyzing the contribution of different innovated modules.
3. The two-stage pretraining strategy is clever, which could serve as a general training techniques for this field.

**Weaknesses**

1. The principle of the SSBIM module is kind of confusing. Why do we need to update the sequence during the encoding of the VAE? First, the objective of VAE is to encode the ground truth data into a compressed space and decode it back, during which the update of sequences seems unnecessary. Second, after equation (6), the updated sequences are not fed into later modules and thus do not influence the encoded H and Z.

2. The illustrations of the energy guidance are kind of unclear. How are the energy defined? Are they originally proposed in this paper, or are there some references for implementing the energy terms? Also, the ablation of energy guidance seems to be missing in the ablation study.

---

> ### Author Rebuttal · Authors · 2026-03-30
>
> We thank the reviewer for highlighting the strengths of our work.
>
> **SSBIM.** We agree that the explanation of this module in the original manuscript lacks clarity. The sequence update in SSBIM does not intend to alter the ground truth sequence but introduces a bidirectional interaction between the sequence and the geometry during the encoding phase. This mechanism allows the geometric representation to integrate sequence information before proceeding to latent variable parameterization, ensuring that the representation is not dominated solely by structural neighborhood propagation. Regarding the impact on $H_i$ and $Z_i$, the latent variables are parameterized by the updated geometric representation ${\hat{\hat{h}}}_i$, which incorporates sequence context through cross-attention. Therefore, the sequence information indirectly enters the latent variable modeling through the geometric branch. The updated sequence representation ${\hat{S}}_i$ does not directly participate in subsequent computations, as its primary role is to maintain the symmetry of the bidirectional interaction.
>
> **Energy guidance $E_{phys}$.** We agree that the original manuscript lacks a clear explanation of the energy guidance, particularly regarding the definition of $E_{phys}$, which potentially hinders comprehension and reproducibility. In our implementation, $E_{phys}$ acts neither as a black-box energy function derived from an external scorer nor as a test-time optimization objective directly obtained by invoking PyRosetta. Instead, it serves as a differentiable constraint term formulated from geometric and physical priors, defined as:
>
> $E_{phys}=E_{consec}+E_{inner}+E_{outer}+E_{angle}+E_{torsion}+E_{radius}$
>
> The term $E_{consec}$ represents the adjacent residue distance constraint, defined as:
>
> $E_{consec}=\sum_{i}{max\left(0,d_i-u\right)+max\left(0,l-d_i\right)}$
>
> where $d_i=\left|\left|z_{i+1}-z_i\right|\right|$, the interval $\left[l,u\right]=\left[\mu_t-k\sigma_t,\mu_t+k\sigma_t\right]$, where $\mu_t$ and $\sigma_t$ denote statistics derived from the training data, and $k$ represents a tolerance coefficient.
> The intra-peptide clash constraint $E_{inner}$ and the receptor clash constraint $E_{outer}$ are defined as:
>
> $E_{inner}=\sum_{\left|i-j\right|>1} m a x\left(0,\mu_t-\left|\left|z_i-z_j\right|\right|\right)$
>
> $E_{outer}=\sum_{i,j} m a x\left(0,\mu_t-\left|\left|z_i-x_j\right|\right|\right)$
>
> The bond angle constraint $E_{angle}$ is defined as:
>
> $E_{angle}=\sum_{i}{(\theta_i-\theta_0)^2}$
>
> where $\theta_0$ serves as a reference angle obtained from data statistics. The dihedral angle constraint $E_{torsion}$ is defined as:
>
> $E_{torsion}=\mathbb{E}\left[max\left(0,\left|\phi_i\right|-\phi_0\right)\right]$
>
> where $\phi_0$ originates from the statistical distribution properties of protein backbone dihedral angles characterized by the Ramachandran plot[1]. The spatial distribution constraint $E_{radius}$ is defined as:
>
> $E_{radius}=(std(||z_i-\bar{z}||)-r_0)^2$
>
> where $r_0$ denotes a scale parameter derived from data statistics.
>
> Furthermore, regarding the observation that the ablation study on energy guidance appears absent, we clarify that the ablation removing the time-dependent energy guidance (denoted as w/o T) already exists in the manuscript. However, the original text fails to explicitly identify this as the removal of energy guidance, resulting in a less intuitive presentation. Taking the suggestion of the reviewer into account, we further clarify this aspect in the revised manuscript and include supplementary comparisons involving a constant $\lambda$ and an ablated $\lambda(t)$. This addition more clearly demonstrates the independent contribution of the energy guidance and the effectiveness of its time scheduling strategy.
>
> **Channel graph.** The term channel refers to the atom-level representation within each residue rather than three specific channels. In our implementation, each residue is represented as $R_i=\{r_{i,1},\ldots,r_{i,C}\}$, where $C$ denotes the number of channels determined by data distribution statistics. We set $C$ to 14, corresponding to a set of predefined atomic positions including backbone and side-chain atoms, with $r_{i,c}\in \mathrm{R}^3$ representing the spatial 3D coordinates.
>
> **Success rate.** The interpretation of the reviewer is correct. In our experiments, we define Success based on the interfacial binding energy ∆G. Specifically, a generated peptide structure is considered successful if its binding energy, as evaluated by PyRosetta, satisfies ∆G < 0, indicating a stable binding tendency. This definition is consistent with existing works such as PepGLAD and follows the standard evaluation protocols for this task, ensuring comparability across different methods.
>
> [1] Zhou A Q, O'Hern C S, Regan L. “Revisiting the Ramachandran plot from a new angle”

---

> > ### Author Rebuttal · Reviewer_MHVF · 2026-04-03
> >
> > Thanks for the responses, which fully solved my concerns.

---

> > > ### Author Response · Authors · 2026-04-04
> > >
> > > Thank you very much for reading our rebuttal and adjusting our score, and we're glad to see that our responses have addressed your concerns.

---

### Official Review · Reviewer_5WHb · 2026-03-11

**Soundness:** 4
**Presentation:** 4
**Significance:** 3
**Originality:** 4
**Overall Recommendation:** 5
**Confidence:** 4

**Summary:**

This paper addresses receptor-conditioned full-atom peptide generation, where existing models struggle to model the strongly coupled sequence–structure relationship while simultaneously ensuring conformational diversity and physical feasibility. It proposes PepFGLD. The method uses Flex-VAE to encode peptide fragments into a latent space and trains a diffusion model in this latent space. During inference, the diffusion outputs are fed into the Flex-VAE decoder to obtain full-atom peptides. The Flex-VAE encoder consists of FlexEGNN, SSBIM, and kernel-based nonlinear mapping. Flex-VAE aims to provide an equivariant structural encoding with explicit sensitivity to conformational flexibility; SSBIM performs bidirectional cross-attention between the sequence stream and the geometric stream. The paper also introduces energy guidance during diffusion-based generation to ensure the physical feasibility of generated results.
Strengths and Weaknesses

**Compliance With Llm Reviewing Policy:**

Affirmed.

**Final Justification:**

Overall, it has a certain degree of novelty and some practical engineering value.

**Key Questions For Authors:**

Can you try training the VAE and diffusion jointly? What are the advantages of separate training compared with joint training?

In SSBIM, are μ and ϑ trainable parameters or fixed hyperparameters?

In TDEG, what is the functional form of λ(t)?

**Limitations:**

yes

**Strengths And Weaknesses:**

Strengths

Strong innovation: It combines a VAE model with a diffusion model and, according to task characteristics, substantially modifies the VAE encoder and introduces energy guidance during diffusion-based generation.

Clear motivation and mechanism: It provides relatively clear qualitative and quantitative analyses (with theoretical support) of the design motivations and the mechanisms of the main innovations.

Weaknesses

Some descriptions are unclear: In SSBIM, it is not stated whether μ and ϑ are trainable parameters or fixed hyperparameters. Although kernel-based nonlinear mapping is cited, it would be better to provide in the main text the iterative functional forms of ϕ and R. In TDEG, the functional form of λ(t) is also not given.

The method is relatively cumbersome: It requires training the VAE first and then training the diffusion model. The VAE includes a graph neural network, a cross-attention mechanism, and kernel-based nonlinear mapping, which may lead to high computational cost.

---

> ### Author Rebuttal · Authors · 2026-03-30
>
> We thank the reviewer for highlighting the strengths of our work.
>
> **Joint training.** Latent diffusion is not first proposed here, as we follow existing paradigms such as PepGLAD. We adopt a two-stage training strategy mainly to ensure latent space stability. Training a diffusion model requires a stable latent space distribution; joint training on an unconverged latent space introduces non-stationary objectives hindering learning and convergence. We consider joint training feasible if consistent objectives, including reconstruction, KL constraints, and denoising losses, enable collaborative optimization of the encoder, decoder, and diffusion model. In contrast, a two-stage approach first learns stable and physically consistent latent representations before generative modeling. This decoupling of representation learning from distribution modeling improves training stability and generation reliability. Following the reviewer's suggestion, we conduct experiments on joint training. Alongside existing Flex-VAE and LDM task losses, we introduce a reconstruction loss $L_{rec}$ to ensure the decoder reconstructs the encoder output during joint training, maintaining consistency between the latent space and original structural semantics. We also incorporate a regularization term $L_{reg}$ to constrain generation and prevent severe shifts in the latent variable distribution during training. Unfortunately, these objectives fail to stabilize the data distribution in joint training. Although the Con reaches 0.9677, the Div is only 0.025, and the Success is merely 0.67%, which indicate that while the model learns the coupling between sequence and structure, mode collapse occurs. Generated results concentrate within a narrow distribution, lacking effective exploration. We consider this a promising optimization direction for future latent diffusion research.
>
> **$\mu$ and $\vartheta$ in SSBIM.** While the original manuscript only states that $\mu$ and $\vartheta$ are scalar gating coefficients, we clarify that both are trainable parameters in our setup.
>
> **$\lambda\left(t\right)$ in TDEG.** The original manuscript briefly describes $\lambda\left(t\right)$ as an increasing schedule that emphasizes the constraint only as the noise level decreases, which admittedly limits reader comprehension. In PepFGLD, the diffusion step $t$ is first normalized to $\tau=\frac{t}{T}$, where $T$ represents the total number of diffusion steps. The schedule $\lambda\left(t\right)$ is defined as:
>
> $\lambda(t)=\lambda_{\min}, \quad \tau < \tau_0$
>
> $\lambda(t)=\lambda_{\min}+(\lambda_{\max}-\lambda_{\min})\cdot\frac{\tau-\tau_0}{1-\tau_0}, \quad \tau \ge \tau_0$
>
> where $\lambda_{min}$ and $\lambda_{max}$ denote the lower and upper bounds of the guidance strength respectively, and $\tau_0$ marks the transition point from the exploration phase to the refinement phase. These variables serve as hyperparameters. We set $\lambda_{min}$ to 0.1, $\lambda_{max}$ to 0.7, and $\tau_0$ to 0.7. This parameter configuration maintains weak guidance during most of the denoising process to preserve exploration capacity and progressively strengthens the physical constraints only in the final stages to improve structural feasibility.
>
> **kernel-based nonlinear mapping.** We agree that the original manuscript lacks explicit definitions for $\varphi$ and $R$. In our implementation, $\psi_H$ and $\psi_Z$ employ a kernel-based nonlinear mapping inspired by the Kolmogorov-Arnold Network, which decomposes the transformation into a global nonlinear component and a local kernel expansion component. Specifically, $\varphi(h)$ in Equation 8 represents a smooth global nonlinear mapping designed to model the overall trend, whereas $R(h,C)=[e^{-|h-c_1|^2/\gamma},...,e^{-|h-c_M|^2/\gamma}]$ denotes a radial basis function expansion based on a set of reference points $C = \{c_1, ..., c_M\}$. This expansion captures local variation patterns within the feature space, with $\gamma$ serving as the bandwidth parameter. The function $\psi_Z$ shares this identical formulation while maintaining independent parameters.
>
> **Computation cost.** We agree that PepFGLD incorporates multiple components, yielding a relatively complex architecture. However, the increase in computational overhead is limited in actual model scale. Using PepGLAD as a baseline with approximately 3.58M parameters, PepFGLD has 4.07M parameters, reflecting an increase of about 13.7%. On an NVIDIA A100-40G GPU, when sampling 40 candidate structures per test sample, the average sampling time per sample is 26.73 seconds for PepFGLD and 19.91 seconds for PepGLAD. These results show that although PepFGLD increases computational overhead to some extent, both methods remain within the same order of magnitude.

---

> > ### Author Rebuttal · Reviewer_5WHb · 2026-04-06
> >
> > The main weaknesses and questions have been addressed.

---

> > > ### Author Response · Authors · 2026-04-06
> > >
> > > We once again appreciate your recognition of our work, and we're glad to see that our responses have addressed your concerns.

---

### Official Review · Reviewer_ejfW · 2026-03-13

**Soundness:** 3
**Presentation:** 3
**Significance:** 3
**Originality:** 2
**Overall Recommendation:** 5
**Confidence:** 2

**Summary:**

This paper proposes PepFGLD, a latent diffusion framework for full atom resolution peptide design with receptor as conditions, explicitly considering the flexibility of the peptide. The method combines three components: (i) FlexEGNNm a flexibility aware equivariant encoder that adapts geometric interactions based on local conformational variability, (ii) SSBIM: a bidirectional sequence-structure coupling in latent space with nonlinear mapping, and (iii) TDEF, a time-dependent energy-guided diffusion mechanism that enforces physical constraints at parts where noise are decreased. Experiments over sequence-structure co-design, binding conformation generation, binding-energy fidelity, and case study are used used to verify the method.

**Compliance With Llm Reviewing Policy:**

Affirmed.

**Final Justification:**

During the rebuttal, the authors haved resolved my concern and question regarding the analysis on the time-dependent energy guidance. The experiments show that two components perform better in terms of Delta F and success, with the cost of others. However, this is not an analysis to reject or wekanes the main claim of the main paper.

Overall, I keep my score, which has bene already learning toward acceptance.

**Key Questions For Authors:**

1. Performance of PepGLAD on PepBench (soundness)

The performance of PepGLAD on PepBench was reported to have a $\Delta G$ of -21.94 and success of 55.97%, while the Table 1 reports it as -11.21 and 47.12%. The authors have mentioned that a small number of entries that failed coordinate parsing were removed from the original datasets, which is described as a minor change. Could the authors clarify what specifically accounts for this performance gap? Though it is denoted as a minor change, it seems like quite a discrepancy.

2. Analysis on the time-dependent energy guidance (TDEG)

The ablation of removing TDEG shows that the performance degrades, but I am also curious of different design choices for the time-dependent schedule $\lambda (t)$ changes the performance. A simple ablation results such as a constant guidance $\lambda$ fixed would solve my worries.

**Limitations:**

yes

**Strengths And Weaknesses:**

**Strengths**

1. Flexibility considered throughout the pipeline (significance)

The core contribution of this paper is that is address flexibility at every level, compared to prior works. When encoding, FlexEGNN incorporates local conformational flexibility, at latent representation SSBIM and nonlinear mapping capture multimodal conformational distributions, and the diffusion process applies physical guidance to balance exploration and convergence towards feasible structures. This design is well-motivated, considering that peptide flexibility manifests across the pipeline, while prior works leave gaps.

2. Comprehensive evaluation (soundness)

Experiments over sequence-structure co-design, binding conformation generation, binding-energy fidelity. Results demonstrate competitive or SOTA performance, particularly in binding energy and success rate. However, there is one question regarding the baselines results of sequence-structure co-design task.

**Weaknesses**

1. Computation cost (soundness)

The proposed framework is consisted of multiple components beyond the PepGLAD style pipeline, FlexEGGN, SSBIM, nonlinear latent mappings, and TDEG with energy gradient computation. While PepGLAD has reported samplings (3 seconds per sampling per candidate, I don’t find any computation cost (time neither space) in the paper. Given the modest improvements, I am curious about the complexity due to the long pipeline.

2. (Minor) Receptor-side flexibility not modeled (soundness)

As the authors have mentioned, the receptor is yet considered flexible. This is a reasonable scope limitation, the frameworks’ emphasis on peptide flexibility may be less effective for targets where receptor rearrangement plays a dominant role in binding. This is noted as a future work, and thus not a major weakness though.

---

> ### Author Rebuttal · Authors · 2026-03-30
>
> We thank the reviewer for highlighting the strengths of our work.
>
> **Computation cost.** We agree that reporting computational overhead statistics helps determine whether the complexity introduced by the longer pipeline is justified. Since the original PepGLAD paper does not specify its hardware and runtime configuration, we re-evaluate both methods under an identical experimental environment to ensure a fair comparison. On an NVIDIA A100-40G GPU, when sampling 40 candidate structures for each test sample, the average sampling time per sample is 26.73 seconds for PepFGLD and 19.91 seconds for PepGLAD. Additionally, the parameter count is approximately 3.58M for PepGLAD and 4.07M for PepFGLD, reflecting an increase of roughly 13.7%. Experimental results demonstrate that the complex architecture of PepFGLD increases the computational overhead and parameter size to a certain extent, yet both remain within the same order of magnitude.
>
> **Receptor-side flexibility not modeled.** The current work employs a receptor-conditioned problem formulation where the receptor serves as a fixed structural context, allowing us to focus on modeling the conformational flexibility of the peptide. In the revised manuscript, we update the limitation discussion to state that a remaining limitation is that receptor-side conformational dynamics are not explicitly modeled, particularly for targets dominated by receptor rearrangements, which is noted as future work.
>
> **Performance of PepGLAD on PepBench.** We agree that stating there are a small number of entries in the current manuscript lacks precision and causes potential misunderstanding, which we correct in the revised version. To clarify, the PepGLAD results reported in Table 1 do not directly originate from the values in the original paper but come from retraining and evaluating the model under an experimental setup identical to that of PepFGLD. We attribute the discrepancy primarily to differences in data processing pipelines. In the original implementation, certain samples may be handled using alternative strategies (e.g., accommodating locally missing atoms), allowing them to remain in the training and evaluation sets. In contrast, we adopt a unified processing pipeline and retain only samples that can be consistently processed throughout the full pipeline. To verify that this process introduces no distribution shift, we compare the data distributions before and after cleaning based on peptide length and calculate a Jensen-Shannon divergence of 0.0071, which indicates high distributional consistency. Therefore, the discrepancy arises mainly from implementation-level processing differences rather than the underlying data or method.
>
> **Analysis on the time-dependent energy guidance (TDEG).** We agree that the original manuscript only reports the results of removing TDEG (w/o T) without investigating the ablation of $\lambda\left(t\right)$. We include a control experiment with a constant $\lambda$=0.5, and the following Table presents the results. Experimental results indicate that, compared to the w/o T setting, energy guidance with a constant $\lambda$ improves performance. Specifically, ∆G improves from -16.77 to -19.15 and Success increases from 53.52% to 55.51%, which confirms the effectiveness of the energy constraint itself. Compared to the constant $\lambda$ configuration, PepFGLD achieves further enhancements in both ∆G and Success, validating the effectiveness of $\lambda\left(t\right)$. The results also reveal that the Con and Div metrics of PepFGLD experience a slight decline compared to the two variants. This occurs because the time schedule progressively strengthens the constraint in the later stages, which concentrates the generation distribution more tightly on physically feasible regions. Consequently, this mechanism partially compresses the exploration space but yields superior binding quality and overall feasibility.
>
> | Model              | $\Delta G \downarrow$ | Success $\uparrow$ | Div $\uparrow$ | Con $\uparrow$ |
> |--------------------|----------------------|--------------------|----------------|----------------|
> | w/o T           | -16.77               | 53.52%             | **0.6623**     | 0.8323         |
> | Constant $\lambda$ | -19.15               | 55.51%             | 0.6193         | **0.8325**     |
> | PepFGLD      | **-19.86**           | **59.35%**         | 0.5824         | 0.8160         |

---

> > ### Author Rebuttal · Reviewer_ejfW · 2026-04-03
> >
> > I thank the author's for the detailed response and ablation experiments.
> > I will keep my score, since already leaning toward acceptance.

---

> > > ### Author Response · Authors · 2026-04-04
> > >
> > > We once again appreciate your recognition of our work, and we're glad to see that our responses have addressed your concerns.

---

### Decision · Program_Chairs · 2026-04-30

**Decision:**

Accept (regular)

**Comment:**

This paper presents a latent diffusion framework (PepFGLD) that effectively tackles the challenge of explicitly modeling conformational flexibility in full-atom peptide design. The reviewers agreed that the method is well-motivated, and the authors provided a strong rebuttal that comprehensively resolved all initial concerns regarding energy guidance formulations, computational overhead, and baseline comparisons.